# Low $p$CO$_2$ under sea-ice melt in the Canada Basin of the western Arctic Ocean

Naohiro Kosugi[1], Daisuke Sasano[2], Masao Ishii[1], Shigeto Nishino[3], Hiroshi Uchida[3], Hisayuki Yoshikawa-Inoue[4]

[1]Meteorological Research Institute, Tsukuba, 305-0052, Japan
[2]Japan Meteorological Agency, Tokyo, 100-8122, Japan
[3]Japan Agency for Marine-Earth Science and Technology, Yokosuka, 237-0061, Japan
[4]Hokkaido University, Sapporo, 060-0810, Japan

*Correspondence to*: Naohiro Kosugi (nkosugi@mri-jma.go.jp)

**Abstract.** In September 2013, we observed an expanse of surface water with low CO$_2$ partial pressure ($p$CO$_2$$^{sea}$) ($< 200$ µatm) in the Chukchi Sea of the western Arctic Ocean. The large undersaturation of CO$_2$ in this region was the result of massive primary production after the sea-ice retreat in June and July. In the surface of the Canada Basin, salinity was low ($< 27$) and $p$CO$_2$$^{sea}$ was closer to the air–sea CO$_2$ equilibrium (~360 µatm). From the relationships between salinity and total alkalinity, we confirmed that the low salinity in the Canada Basin was due to the larger fraction of meltwater input (~0.16) rather than the riverine discharge (~0.1). Such an increase in $p$CO$_2$$^{sea}$ was not so clear in the coastal region near Point Barrow, where the fraction of riverine discharge was larger than that of sea-ice melt. We also identified low $p$CO$_2$$^{sea}$ ($< 250$ µatm) in the depth of 30-50 m under the halocline of the Canada Basin. This subsurface low $p$CO$_2$$^{sea}$ was attributed to the advection of Pacific-origin water, in which DIC is relatively low, through the Chukchi Sea where net primary production is high. Oxygen supersaturation ($> 20$ µmol kg$^{-1}$) in the subsurface low $p$CO$_2$$^{sea}$ layer in the Canada Basin indicated significant net primary production undersea and/or in preformed condition. If these low $p$CO$_2$$^{sea}$ layers surface by wind mixing, they will act as additional CO$_2$ sinks; however, this is unlikely because intensification of stratification by sea-ice melt inhibits mixing across the halocline.

## 1 Introduction

The extent of sea ice and its thickness in the Arctic Ocean have been declining in recent decades (Comiso, 2012; Stroeve et al., 2012a, 2012b); these declines are widely considered a consequence of climate change resulting from the emissions of anthropogenic greenhouse gases. The average monthly extent of sea ice in September in the Arctic Ocean decreased by about 12.4% per decade from 1979 to 2010 (Stroeve et al., 2012b). In September 2012, the area of sea ice in the Arctic Ocean was less than $4 \times 10^6$ km$^2$, about 50% of the average in the 1980s (Parkinson and Comiso, 2013). Because of this decline in the extent of sea ice, the air–sea CO$_2$ flux in the Arctic Ocean is also thought to be dramatically changing. Currently, the Arctic CO$_2$ sink has been estimated as 66–199 Tg C yr$^{-1}$, with a large uncertainty (Bates and Mathis, 2009; Yasunaka et al., 2016). This value is equivalent to 3–8% of the net CO$_2$ sink of the global ocean ($2.6 \pm 0.5$ Pg C yr$^{-1}$ in the

period 2006–2015; Le Quéré et al., 2016). A recent modeling study suggests that the $CO_2$ sink in the Arctic Ocean is increasing (Manizza et al., 2013).

The reasons for an increasing $CO_2$ sink in the Arctic Ocean include the increase in the area and duration of ice-free conditions and the enhanced net primary production they induce. However, the effect of increasing meltwater input that accompanies the sea ice decline should also be taken into consideration (Yamamoto-Kawai et al., 2009; Rabe et al., 2014). The large input of ice melt enhances stratification in the upper layer and forms a thin surface mixed layer with a distinct halocline below. Although dilution of the surface water with meltwater lowers the partial pressure of $CO_2$ ($pCO_2^{sea}$), shoaling of the surface mixed layer would accelerate equilibration of the surface water with the overlying air. The input of meltwater is also likely to influence carbonate chemistry by altering the ratio of dissolved inorganic carbon (DIC) to total alkalinity (TA), although it is still unclear whether the addition of meltwater increases or decreases the DIC/TA ratio (Rysgaard et al., 2007; Bates et al., 2014). Cai et al. (2010) reported unprecedented high $pCO_2^{sea}$ (~370 µatm) in the Canada Basin in summer. They ascribed this high $pCO_2^{sea}$ to low net primary production and rapid equilibration with atmospheric $CO_2$ in the shallow mixed layer derived from meltwater input. The low nutrient concentration in meltwater limits the biological drawdown of $pCO_2^{sea}$. Else et al. (2013) found that surface warming also contributed significantly to $pCO_2^{sea}$ increase in a shallow mixed layer in the Canada Basin. Both studies concluded that an increase in meltwater lowers $CO_2$ absorbing capacity in the Canada Basin.

A notable feature of the Canada Basin in summer is a complex water-column structure. Because of the strong salinity gradient, there are several maxima and minima of temperature within 150 m of the surface. This water-column structure does not always remain stable in the rapidly changing Arctic Ocean. Although Cai et al. (2010) and Else et al. (2013) reported relatively high surface $pCO_2^{sea}$ in the Canada Basin, they did not fully explain the $CO_2$ chemistry below the surface mixed layer there. We studied the water-column $CO_2$ variation in the western Arctic Ocean and the processes that cause it.

In late summer 2013, we made shipboard observations in the Chukchi Sea and in the Canada Basin of the western Arctic Ocean. While underway or at hydrographic stations, or both, we measured temperature, salinity, dissolved oxygen, and the carbonate system variables $pCO_2^{sea}$, DIC, and TA. From the salinity–TA relationship, we also mapped the mixing ratio of sea-ice meltwater, riverine outflow, and water of Pacific origin that entered into the western Arctic Ocean through the Bering Strait. The results demonstrate the importance of large net primary production in reducing the $pCO_2^{sea}$ and increasing air-to-sea $CO_2$ flux in the Chukchi Sea. Although this low $pCO_2^{sea}$ water is advected into the Canada Basin, air-to-sea $CO_2$ flux there is blocked by a stratified shallow surface layer that is formed by a large ice-melt input.

## 2. Measurements and Data

Oceanographic measurements in the Chukchi Sea and the Canada Basin were made during cruise MR13-06 of the R/V *Mirai* conducted by the Japan Agency for Marine-Earth Science and Technology from 28 August to 6 October 2013 (Nishino et al., 2015). The port of departure and arrival was Dutch Harbor, Alaska, USA. Most of data used in this paper are

available from the JAMSTEC Data Site for Research Cruises (http://www.godac.jamstec.go.jp/darwin/cruise/mirai/mr13-06_leg1/e). The rest of the data will be opened as soon as they are ready.

We made underway measurements of $p\mathrm{CO_2}^{\mathrm{sea}}$ and DIC together with temperature ($T$) and salinity ($S$) in seawater pumped continuously from an intake located 4.5 m below surface. For the measurement of $p\mathrm{CO_2}^{\mathrm{sea}}$, the pumped water was continuously introduced into a shower-type equilibrator with 1.4 dm$^3$ headspace at a rate of 4 dm$^3$ min$^{-1}$. A wavelength-scanned cavity ring-down spectrometer (Picarro G2301, Picarro Inc., USA) was used to measure the concentrations of $CO_2$ in the headspace of the equilibrator and in the atmosphere sampled from the foremast. The instrument was stable and calibrated once a day against a set of three working standard gases of $CO_2$ in air (Japan Fine Products, 206.34 to 489.28 ppmv) that had been standardized on the WMO X2007 scale (Zhao and Tans, 2006). Response of the instrument to the $CO_2$ concentration was linear. The residual of each standard gas from liner regression was less than 0.03 ppmv. According to the manufacturer's report, precision of $CO_2$ measurement above 500 ppmv is 0.1%.

For the underway measurement of DIC, a portion of pumped water was automatically taken every 15 minutes and filled  into glass bottles (SCHOTT DURAN®; 300 cm$^3$) that have been capped with a screw type lid. Filling, transport and discharge of the samples were all done through high-density PFA-tubes mounted through the lid. DIC was measured after the temperature of the sample seawater was adjusted to 20.0 °C in a thermostated water bath for 1 hour, using a $CO_2$ extraction/coulometric titration system (Nippon ANS, Inc., Japan). This system was comprised of seawater dispensing unit, a $CO_2$ extraction unit, and a coulometer (Model 3000, Nippon ANS, Inc., Japan). This unit dispenses the seawater from sample bottle to a water-jacketed glass pipette of nominal 15 cm$^3$ volume. The temperature of seawater samples in the pipette was kept at 20.0 °C by a water jacket. The sample was then transferred into a glass stripping chamber and stripped of DIC by adding 2 cm$^3$ of phosphoric acid (10% v/v). The emerged $CO_2$ was extracted into the stream of nitrogen gas (130 cm$^3$ min$^{-1}$) and transferred to the coulometer. The system was standardized with Certified Reference Material (CRM; Batch #113) supplied by A. G. Dickson (Scripps Institution of Oceanography); underway measurement of DIC was interrupted for several hours once a day for calibration. The precision of measurement determined by repeatability of CRMs was ±2.2 μmol kg$^{-1}$. The values of TA in the surface were calculated from measured $p\mathrm{CO_2}^{\mathrm{sea}}$, DIC, temperature and salinity using dissociation constants of carbonic acid given by Lueker et al. (2000) and macro package of CO2SYS program for Microsoft Excel (Pierrot et al., 2006). Uncertainty in surface TA was estimated as ±3 μmol kg$^{-1}$ by taking the uncertainties of DIC and $p\mathrm{CO_2}^{\mathrm{sea}}$ into account.

At hydrographic stations, seawater-column profiles of temperature, salinity, and dissolved oxygen were obtained with a CTD (SBE 9 plus, Sea-Bird Scientific, USA) rosette sampler equipped with sensors for dissolved oxygen (SBE 43, Sea-Bird Scientific, USA) and Niskin bottles (12 dm$^3$). In addition to the CTD casts, some expendable CTDs (XCTD) were used to obtain water-column profiles of temperature and salinity. Discrete water samples were taken into Niskin bottles  at depths of 0, 5, 10, 20, 30, 40, 50, 75, 100, 125, 150 and 200 m along with CTD measurements. Samples were also collected at chlorophyll maximum layers that ranged from 12 m to 92 m. Measurements of dissolved oxygen were made by the Winkler titration method following Dickson (1994) and used to correct for the bias of the data from the oxygen sensor SBE

43. Apparent oxygen utilization (AOU), *i.e.*, the difference between the measured concentration of dissolved oxygen and its saturation concentration under the same potential temperature and salinity conditions, was calculated using the oxygen solubility constant given by Garcia and Gordon (1992). Water samples for chlorophyll a were vacuum-filtered ($< 0.02$ MPa) through a 25-mm-diameter Whatman grade GF/F filter, and fluorescence was measured for each sample with a fluorometer (10-AU-005, Turner Designs, USA). The fluorometer was calibrated against pure chlorophyll-*a* (Sigma-Aldrich, USA). The concentration of nutrients (nitrate, nitrite, silicate, phosphate and ammonia) was determined using a continuous flow analyzer (QuAAtro 2-HR, BLTEC, Japan) according to the GO-SHIP Repeat Hydrography Manual (Hydes et al., 2010). Reference materials for nutrients of seawater (Aoyama and Hydes, 2010) were used.

Subsamples for DIC and TA measurements in the discrete water samples were drawn into borosilicate glass bottles (300 cm$^3$ for DIC and 125 cm$^3$ for TA) using the protocol of Dickson et al. (2007). Measurements of DIC at depths were also made with the extraction/coulometric system (Nippon Ans., Japan). Saturated solution of $HgCl_2$ (0.1 cm$^3$) was added to each of the samples to inhibit any biological activity. Measurements of TA were made with a spectrophotometric system (Nippon Ans., Japan) based on a single-point pH determination using bromocresol green as an indicator dye (Yao and Byrne, 1998). Seawater sampled in the glass bottles was transferred to an optical cell via dispensing unit. The length and volume of the cell were 8 cm and 13 cm$^3$, respectively. Temperature of the cell was kept at 25.0 °C. Sample was mixed for 8.5 minutes after the injecting the indicator dye solution and hydrochlonic acid (0.05M). TA was calculated from absorbance ratio at 444 and 616 nm ($A_{616}/A_{444}$). Replicate measurements yielded an average and standard deviation of differences of $1.0 \pm 1.1$ μmol kg$^{-1}$ for DIC and $0.9 \pm 0.8$ μmol kg$^{-1}$ for TA. Values of $pCO_2^{sea}$ in discrete water samples were calculated from DIC, TA, temperature, and salinity using dissociation constants of carbonic acid given by Lueker et al. (2000).

Satellite-derived net primary production (NPP) was used to evaluate biological production in a broad area. NPP is estimated from empirical equations using chlorophyll concentration, sea surface temperature, photosynthetically active radiation and length of the daytime as variables in Vertical Generalized Production Model (Bahrenfeld and Falkowski, 1997). The data of NPP was downloaded from website of Oregon State University (http://www.science.oregonstate.edu/ocean.productivity/standard.product.php).

Monthly averaged wind speed data from the Japanese 55 year reanalysis (JRA-55) product was used to calculate air-sea $CO_2$ flux. JRA-55 has a spatial resolution of 1.25° longitude by 1.25° latitude (Kobayashi et al., 2015). We applied data of reanalysis rather than those of an anemometer mounted on the ship considering the representativeness of data.

## 3. Calculations

### 3.1 $CO_2$ flux and half-life of $\Delta pCO_2$

Surface $pCO_2^{sea}$ was calculated from the mole fraction of $CO_2$ in the air by taking the water vapor pressure and atmospheric pressure into account. The temperature and salinity of the pumped water at the intake were continuously measured with sensors SBE 38 and SBE 45, respectively (Sea-Bird Electronics, USA). PT100 thermometer was equipped

with the equilibrator. An increase in temperature between in situ seawater ($T_{in}$) and the equilibrator ($T_{eq}$) was typically about 0.2 °C. Equation (1) in Takahashi et al. (2009) was applied to convert $pCO_2(T_{eq})$ to $pCO_2(T_{in})$. Uncertainty in the value of $pCO_2^{sea}$ measured using the same type of equilibrator has been estimated to be ±3 μatm (Midorikawa et al., 2006).

Air–sea $CO_2$ flux ($F_{CO2}$) was calculated using the following equation:

$$\Delta pCO_2 = pCO_2^{sea} - pCO_2^{air} \qquad (1)$$

$$F_{CO2} = k\alpha\Delta pCO_2 \qquad (2)$$

where α denotes the solubility of $CO_2$ in seawater (Weiss 1974). We used a gas-transfer piston velocity $k$ given by Wanninkhof (2014):

$$k = 0.251 \times U_{10}^2 \times (Sc/660)^{-0.5} \qquad (3)$$

$U_{10}$, and $Sc$ denote wind speed at 10 m above sea level, and the Schmidt number (Wanninkhof, 2014), respectively.

Half-life of $CO_2$ gas exchange ($\tau_{1/2}$) was calculated for quantitative discussion about temporal variation in $\Delta pCO_2$. At first, initial condition of temperature, salinity, DIC and TA were set. Initial $pCO_2$ ($pCO_2^0$) was calculated from these values using dissociation constants of carbonic acid given by Lueker et al. (2000).

$$pCO_2^0 = f(T, S, DIC, TA) \qquad (4)$$

All parameters but DIC were fixed during the calculation, i.e., temperature, salinity and TA were assumed unchanged. $F_{CO2}$ in each time step was calculated using equation (2), (3) and (4). Increase in DIC was calculated from $F_{CO2}$. Time step was set to one day.

$$\Delta DIC = \frac{F_{CO2}}{MLD * \rho(T,S)} \qquad (5)$$

$$DIC_{t+1} = DIC_t + \Delta DIC \qquad (6)$$

Here, MLD and (T, S) mean fixed mixed layer depth [m] and density of seawater in surface mixed layer [kg m$^{-3}$] respectively. After each time step, $\Delta pCO_2^t$ and $pCO_2^t$ were calculated from DIC. Time required to reduce $\Delta pCO_2^t$ to half of $\Delta pCO_2^0$ was defined as $\tau_{1/2}$.

**3.2 Determination of freshwater fraction**

In the western Arctic Ocean, the water at the temperature minimum layer (~150 dbar) is known to originate in the North Pacific and be advected into the Arctic through the Bering Strait (POW: Pacific Origin Water, Shimada et al., 2005), and the water in the layer above the temperature minimum is thought to be a mixture of this POW with sea-ice melt and riverine outflow. To determine the fractions (*f*) of the three different source waters — POW, sea-ice melt (SIM), and riverine outflow (RRO) — in the upper-layer waters of the Chukchi Sea and the Canada Basin, we used the relationship between salinity and TA and the following mass balance equations.

$$TA = f_{POW} \cdot TA_{POW} + f_{SIM} \cdot TA_{SIM} + f_{RRO} \cdot TA_{RRO} \qquad (7)$$

$$S = f_{POW} \cdot S_{POW} + f_{SIM} \cdot S_{SIM} + f_{RRO} \cdot S_{RRO} \qquad (8)$$

$$1 = f_{POW} + f_{SIM} + f_{RRO} \qquad (9)$$

We chose the data of TA and $S$ from 38 sampling layers/locations in the temperature minimum layer where $T < -1.5$ °C in the Chukchi Sea and the Canada Basin during the cruise, and defined their means ($2264.2 \pm 12.6$ µmol kg$^{-1}$ and $32.89 \pm 0.22$) as the values of $TA_{POW}$ and $S_{POW}$, respectively (Fig. 1).

There are several studies of TA of riverine outflow in the Arctic. Cooper et al. (2008) directly measured TA in six major rivers in the Arctic: they concluded that flow-weighted average of TA of these six rivers was 1048 µmol kg$^{-1}$. Yamamoto-Kawai et al. (2009) made linear regression analysis of salinity and TA, and reported that the intercept ($S = 0$) was 793 µmol kg$^{-1}$ for the whole Canada Basin. Data of TA and salinity taken during our cruise indicate that the upper limit of distribution in salinity-TA plots (Fig. 2) is consistent with the line extended to this intercept deduced by Yamamoto-Kawai et al. (2009); consequently, we regarded this value as $TA_{RRO}$. In consideration of the spatial and temporal fluctuation of riverine TA, we assumed that the uncertainty of $TA_{RRO}$ is $\pm100$ µmol kg$^{-1}$ (Yamamoto-Kawai et al., 2005). Accordingly, the estimated errors of $f_{SIM}$ and $f_{RRO}$ are as large as $\pm0.02$.

Conversely, values of $S_{SIM}$ and $TA_{SIM}$ so far reported fall within a relatively narrow range. We applied $S_{SIM} = 5$ and $TA_{SIM} = 349$ µmol kg$^{-1}$ following Fransson et al. (2009). Differences in $f_{SIM}$ and $f_{RRO}$ are not larger than $\pm0.008$ when applying other values suggested by Anderson et al. (2004) ($S_{SIM} = 4$, $TA_{SIM} = 263$ µmol kg$^{-1}$). Cumulative error in $f_{SIM}$ and $f_{RRO}$ associated with the selection of the end-member salinity and TA are within $\pm0.03$. As shown in Fig. 2, $S$–TA plots for the Chukchi Sea and Canada Basin fall among the three $S$–TA end-members of POW, SIM, and RRO. Surface water in the Chukchi Sea and Canada Basin consists mainly of POW, but includes sizable $f_{SIM}$ up to 0.16 and $f_{RRO}$ up to 0.18.

## 4. Results and Discussion

### 4.1. Variations in Temperature and Salinity in the Surface Layer

Cruise MR13-06 occupied a wide area of the Chukchi Sea and the Canada Basin. General variations in surface $pCO_2^{sea}$ in these regions have already been well investigated (Bates 2006, Cai et al., 2010). The results from our cruise were not much different from these reports. Therefore, we highlighted below the differences of water mass characteristic and $CO_2$ dymnamics in these regions. In the period from 4 to 11 September 2013, temperature (SST), salinity (SSS), $pCO_2^{sea}$, and DIC in surface water were highly variable in the western Arctic Ocean (Fig. 3), particularly around the continental slope (200 m isodepth in Fig. 3) of the Chukchi Sea. Average $pCO_2^{air}$ measured onboard was 385.0 µatm, which is consistent with the value observed at Point Barrow, Alaska (http://ds.data.jma.go.jp/gmd/wdcgg/). According to JRA-55, average $U_{10}$ in the region north of 70°N was 4-5 m s$^{-1}$ in September 2013.

Variation in SST and SSS on the cruise track was abrupt rather than gradual (Fig. 3a and 3b). Therefore, we defined three subregions; (1) Barrow Coastal Water (BCW), (2) Canada Basin Water (CBW) and (3) Chukchi Sea Water (CSW). The boundary between BCW and CBW was 2°C isotherm at 72.5°N, 154.8°W. CBW and CSW were separated by the 28 psu isohaline at 73.3°N, 168.3°W (Fig. 3c).

The fraction of freshwater had distinct spatial variations among the three subregions (Fig. 3d and 3e; summarized in Table 1). Low salinity in BCW was mainly due to riverine outflow: in this subregion, $f_{RRO}$ was as large as 0.18, presumably because the Alaskan coastal current which flows northward along the Alaskan coast toward Point Barrow contains a considerable fraction of Yukon River outflow (Steele et al., 2004). In contrast, the lower salinity in the CBW was primarily due to the input of meltwater from sea ice, although it also contained significant riverine outflow. In the northernmost region of the Canada Basin (north of 74°N), $f_{SIM}$ was as large as 0.16, whereas $f_{RRO}$ was no more than 0.10 and was almost always lower than $f_{SIM}$. CSW was largely composed of Pacific water and rarely contained riverine outflow as it flowed directly from the Bering Strait.

## 4.2. Variations in carbonate chemistry in the Surface Layer

Remarkable differences in $pCO_2^{sea}$, DIC, and TA were observed among the three subregions (Table 1, Fig. 3f, g, and h). We attributed the low DIC/TA ratio and the low $pCO_2^{sea}$ (< 200 µatm) in CSW to the massive biological activity there in early summer. In this region, net primary production decreases $pCO_2^{sea}$ to 200 µatm or less in early summer (Bates 2006). According to the analysis of satellite imagery, net primary productivity (NPP) in July 2013 was as high as 1000 mgC $m^{-2}$ $day^{-1}$ in the majority of the Chukchi Sea. Even though NPP had decreased to ~500 mgC $m^{-2}$ $day^{-1}$ in September 2013 during our measurements, $pCO_2^{sea}$ had been notably lower than $pCO_2^{air}$ for months after the massive primary production in early summer. This was because of both a still relatively high biological production and slow net $CO_2$ exchange. Under typical summer conditions ($T = 3$ °C, $S = 32$, DIC = 2000 µmol $kg^{-1}$, TA = 2220 µmol $kg^{-1}$, mixed layer depth = 15 m, and $U_{10} = 5.0$ m $s^{-1}$), $\tau_{1/2}$ is considered to be longer than 100 days.

The DIC/TA ratio in CBW was higher than that in CSW (Table 1). The value of $pCO_2^{sea}$ in CBW ranged from 300 to 360 µatm (Fig. 3f). Although the level of $pCO_2^{sea}$ in CBW was still lower than the $pCO_2^{air}$ (385 µatm), it was much higher than that in CSW. The primary cause of the $pCO_2^{sea}$ being nearly as high as $pCO_2^{air}$ is that the addition of meltwater to the surface layer shoals the mixed layer (Cai et al., 2010; Else et al., 2013), thereby reducing the time for surface water to reach air–sea $CO_2$ equilibrium. An additional cause of higher $pCO_2^{sea}$ in the high $f_{SIM}$ region is probably low net primary production, because the concentrations of nutrients in meltwater are low; e.g., Lee et al. (2012) reported that the concentration of nitrate in a melt pond being formed on the top of sea ice in the Canada Basin was low (< 0.5 µM), and that the low nitrate concentration limited biological production in the pond. Our results corroborate previous reports by Cai et al. (2010) and Else et al. (2013) that the overspreading of the surface layer by sea-ice melt inhibits $CO_2$ uptake by the ocean. Near equilibrium $pCO_2^{sea}$ conditions after seasonal sea-ice retreat is likely to be common in the Canada Basin. The impact of sea-ice melt itself on $pCO_2^{sea}$ was difficult to resolve only from our observations. Bates et al. (2014) found both basic (i.e., DIC/TA < 1) and relatively acidic (i.e., DIC/TA > 1) melt ponds in the Canada Basin. To study the impact of meltwater on carbonate chemistry, direct sampling of sea ice into gastight bags (Fransson et al., 2013) will be required.

In BCW, $pCO_2^{sea}$ was about 270 µatm on average, between that in CSW and CBW (Fig. 3f). The fraction of freshwater indicates that surface freshening in BCW is mainly caused by riverine outflow ($f_{RRO} = 0.11$) rather than sea-ice

melt ($f_{SIM}$ = 0.08). Riverine outflow had a higher TA/$S$ ratio than sea-ice melt (Fig. 2). It also has larger content of DIC (Ulfsbo et al., 2014). In our measurements, surface chlorophyll *a* was higher in BCW (0.4 to 2.0 mg dm$^{-3}$) than in CBW (0.1 to 0.3 mg dm$^{-3}$), implying that biological drawdown of DIC was greater in BCW. Consequently, both DIC/TA and $p$CO$_2$$^{sea}$ in BCW were lower than those in CBW. At the time of our observation, BCW still could absorb more CO$_2$ from the atmosphere than offshore CBW. This is an important finding, because river water inflow into the Arctic Ocean is considered highly likely to increase with climate change (McClelland et al., 2006; Déry et al., 2009).

### 4.3. Variations in the Water Column

Water properties differed not only in the surface but also in the water column among these three. *T-S* diagrams obtained by CTD for each subregion are shown in Fig. 4. The surface around Point Barrow was fresh and warm; as depth increases, the water column gradually cooled to the coldest water around $S$ = 33 (Fig. 4a). A similar decrease in temperature from near surface to bottom was observed in the Chukchi Sea (Fig. 4b). In contrast, the water column in the Canada Basin was more complex, with a number of temperature maxima and minima (Fig. 4c). Jackson et al. (2010) classified the water column in the Canada Basin from the top to the bottom into a surface mixed layer, a near surface temperature maximum (NSTM), a remnant of the Winter Mixed Layer (rWML), Pacific Summer Water (PSW), and Pacific Winter Water (PWW). The surface mixed layer had the lowest salinity ($S$ < 27) because almost all sea-ice melt is trapped in this layer during summer. The NSTM is separated from the surface mixed layer by stratification and warmed by the input of solar radiation. The depths of NSTM ranged between 15 and 26 m during our observations. Below the NSTM, the rWML corresponded to the temperature minimum ($T \approx -1$ ℃, $S \approx 29.3$), which was formed in the Canada Basin during the previous winter. Another temperature maximum around $S$ = 30.5 corresponded to PSW, which was advected and modified in the Chukchi Sea during summer. The lowest temperature observed was near the freezing point in PWW at around $S$ = 33.1.

Temperature and salinity were frequently measured along the cruise track by CTD and XCTD sensors. Water-column profiles showed a distinct halocline from 10 to 20 dbar in BCW and CBW (Fig. 5a and 5b). In these two subregions, the difference in salinity between above and below the halocline was up to 2. Unlike the other two subregions, thermocline was more prominent than halocline in CSW. Column variation of $f_{SIM}$ and $f_{RRO}$ indicate that both sea-ice meltwater and river outflow greatly contributed to the formation of the halocline (Fig. 5c and 5d). In the Canada Basin, $f_{SIM}$ was as high as 0.12 ± 0.01 (± standard deviation) in the top layer down to 10 dbar, but decreased abruptly with depth to practically zero (0.01 ± 0.01) in the 29–50 dbar layer. Likewise, $f_{RRO}$ was also quite high (0.09 ± 0.01) in the top layer down to 10 dbar and decreased gradually to 0.06 ± 0.02 in the 29–50 dbar layer. This vertical structure indicates that the input of sea-ice melt occurred shortly before the measurement (at least, in summer 2013) and contributed to the formation of a discrete layer in the surface over the main water mass in the Canada Basin, whereas river outflow had undergone vertical mixing in the course of advection before it reached the Canada Basin.

Among these three subregions, differences were also evident in column $p$CO$_2$$^{sea}$. In the upper layer (above 10 dbar) of CSW and BCW, average $p$CO$_2$$^{sea}$ was 195 ± 11 µatm and 258 ± 14 µatm, respectively. As mentioned in section 4.2, these

low $p$CO$_2$$^{sea}$ values were the result of net primary production. In these subregions, $p$CO$_2$$^{sea}$ increased with depth below the halocline. The water-column profile of AOU indicates that the increase in $p$CO$_2$$^{sea}$ was due to the input of CO$_2$ associated with the degradation of organic matter (Fig. 5e and 5f).

Unlike the water-column profiles of $p$CO$_2$$^{sea}$ and AOU in CSW and BCW, those in CBW were distinctive in that they had subsurface minima. In the top 10 dbar of CBW, $p$CO$_2$$^{sea}$ reached 322 ± 20 µatm, a value still lower than $p$CO$_2$$^{air}$ (~385 µatm) but the highest among the three subregions. However, $p$CO$_2$$^{sea}$ decreased with depth below the halocline and reached 271 ± 31 µatm in the range of $29.3 < S < 31.3$ (30 to 50 dbar layer; Fig. 5e and 6a). Below the halocline in CBW, AOU was significantly negative ($< -20$ µmol kg$^{-1}$) like that in the CSW where net primary production was large (Fig. 5f and 6b). Subsurface maximum of chlorophyll $a$ and dissolved oxygen have also been found in the Canadian Archipelago (Martin et al., 2010). Here, our frequent observations facilitated classification of the water masses and their origins in the Canada Basin. According to the salinity–AOU profile (Figure 6b) in the Canada Basin, AOU was largely negative ($< -20$ µmol kg$^{-1}$) in the salinity range of 28 to 31.5 that corresponds to NSTM, rWML and PSW. NSTW and rWML were formed in the Canada Basin. Nitrates were almost depleted ($< 0.2$ µmol kg$^{-1}$) in these two layers during our observations (Figure 6c). Even in spring, concentration of nitrate was low ($< 2$ µmol kg$^{-1}$) in surface water of the Canada Basin (Codispoti et al., 2005). Sunlight surely reaches the 50 m depth in the Canada Basin although it is not strong (Jackson et al., 2010). Negative AOU in NSTM and rWML indicated the large biological production utilizing the nutrients and the sunlight. Excess oxygen produced by the biological production remained in subsurface as it was isolated from surface by strong halocline. Results of our measurements also showed that significant nutrients remained in PSW where $S > 29.3$ (Figure 6c). Oversaturation of oxygen in PSW was due to the remnant of massive biological production in early summer and/or the production undersea during advection from the Chukchi Sea to the Canada Basin.

To compare the water properties among layers, we calculated preformed nDIC$_{32}$ as defined by the following equation.

$$\text{Preformed nDIC}_{32} = \frac{\text{DIC} - \text{AOU} \times r_{C:O}}{S} \times 32 \qquad (10)$$

Here, $r_{C:O}$ denotes stoichiometric ratio of DIC to AOU being set to 117/170 (Anderson and Sarminento, 1994). As shown in Fig. 6d, preformed nDIC$_{32}$ was almost constant and the highest in the water column from surface to the depth where salinity was 29.3. This indicates that the water above the depth where salinity was 29.3 had the same origin in the Canada Basin. There was no clear minimum of $p$CO$_2$$^{sea}$ in NSTM and rWML in spite of the negative AOU and biological production. . This suggests that the $p$CO$_2$$^{sea}$ minimum in the PSW below was explained by the drawdown of DIC due to biological production when the PSW was in the surface of the Chukchi Sea, in addition to the undersea DIC drawdown. In fact, preformed nDIC$_{32}$ in the salinity range of 29.3 to 33.1 that corresponded to PSW and PWW from the Chukchi Sea was about lower by about 100 µmol kg$^{-1}$ than in the upper layers from Canada Basin (Fig. 6d). However, we have to note that the conventional salinity-normalization like those Eq. (10) overestimates the nDIC in a source water when it is diluted with river

run-off and/or sea-ice melt that contain DIC (Friis et al., 2013). In more strict sense, preformed DIC in the subsurface waters of the Canada Basin is approximated by

$$\text{preformed DIC} = f_{POW} \cdot \text{preformed DIC}_{POW} + f_{SIM} \cdot \text{DIC}_{SIM} + f_{RRO} \cdot \text{DIC}_{RRO} \quad (11)$$

Preformed $DIC_{POW}$ was determined to be 2170 μmol kg$^{-1}$ from salinity, DIC and AOU in temperature minimum layer (PWW). $DIC_{SIM}$ and $DIC_{RRO}$ were assumed to be 300 μmol kg$^{-1}$ (Fransson et al., 2013) and 800 μmol kg$^{-1}$ (Tank et al., 2012), respectively. For the PSW in which salinity range of 29.3 to 31.5, $f_{SIM} = 0.01$ and $f_{RRO} = 0.06$, preformed DIC observed was 2027 μmol kg$^{-1}$. This was lower by 34 μmol kg$^{-1}$ than that calculated from Eq. (11) (2061 μmol kg$^{-1}$), which suggests the biological DIC drawdown in the preformed condition of the PSW. However, for rWML and shallower in which $S < 29.3$, $f_{SIM} = 0.12$ and $f_{RRO} = 0.09$, preformed DIC observed was 1850 μmol kg$^{-1}$. This was consistent with that calculated from Eq. (11) (1858 μmol kg$^{-1}$). These results allowed us to support to conclude that $pCO_2^{sea}$ minimum in subsurface in the Canada Basin was attributable not only to large biological drawdown of DIC but also to lower DIC in PSW as compared with Canada Basin origin water lying above.

## 4.4. Future direction of hidden $CO_2$ sink in the Canada Basin

How the long-term retreat of sea ice changes the air–sea $CO_2$ flux in the Arctic Ocean is a matter of controversy. Manizza et al. (2013) argued that increasing SST will enhance biological primary production and drawdown of $CO_2$ in seawater. Laruelle et al. (2014) asserted that larger ice-free areas and longer ice-free periods will provide greater occasion for oceanic $CO_2$ uptake. In contrast, Cai et al. (2010) and Else et al. (2013) insisted that the increase in sea-ice melt results in the formation of thin surface mixed layers and limits further uptake of $CO_2$ from the atmosphere by this layer.

As a result of our observations that a subsurface minimum of $pCO_2^{sea}$ existed in the Canada Basin, it is necessary to study whether the surface mixed layer there will deepen under a warming climate. If the surface layer is stirred by strong wind and mixed with the subsurface low $pCO_2^{sea}$ layer, the surface will act as a further $CO_2$ sink. Several reports indicate that the strong wind associated with the passage of low pressure systems deepens the surface mixed layer and has impacts on the biogeochemistry (Wada et al., 2011; Rumyantseva et al., 2015). Simmonds and Keay (2009) reported that the strength of cyclones in the Arctic Ocean is increasing with the long-term reduction of sea-ice cover. However, we also have to consider the strength of stratification in the Canada Basin. In a comprehensive analysis of mixed layer depth in the Arctic Ocean, Peralta-Ferriz et al. (2015) found a significant positive correlation between the mixed layer depth and the maximum wind speed in the preceding 5 days ($4.6 \pm 0.8$ m per m sec$^{-1}$) in the case that the differences in density between the mixed layer and 20 m below ($\Delta\rho$) is smaller than 0.5 kg m$^{-3}$. However, in the case of $\Delta\rho > 0.5$ kg m$^{-3}$, deepening of the mixed layer is much less sensitive to the increase in wind speed ($0.77 \pm 0.52$ m per m sec$^{-1}$). In our observations, $\Delta\rho$ exceeded 2.0 kg m$^{-3}$ at all CTD stations in the Canada Basin (Fig. 4c). Hence, we suggest that additional $CO_2$ uptake in the Canada Basin by wind mixing is unlikely because stratification was strong even in 2013 and will be further strengthened by the additional input of sea-ice melt in the future.

Climate change also affects the subsurface layer in the Canada Basin, where low $p\mathrm{CO_2}^{\mathrm{sea}}$ is caused by net primary production. McLaughlin and Carmack (2010) reported that increase in sea-ice melt and the strengthening of Ekman pumping deepened the nutricline and the depth of chlorophyll maximum in the Canada Basin. Nishino et al. (2013) also observed decreases in nitrate and chlorophyll in the 0 to 50 m depth layer in the Canada Basin during 2002–2010; they attributed these decreases to the decrease in inflow of nutrient-rich water from the East Siberia Sea. In either case, biological production below the halocline of the Canada Basin is likely to decrease in the long term. In this regard, it seems unlikely that the subsurface low $p\mathrm{CO_2}^{\mathrm{sea}}$ layer in the Canada Basin will act as another $\mathrm{CO_2}$ sink.

## 5. Conclusions

A wide range of surface $p\mathrm{CO_2}^{\mathrm{sea}}$ was observed in the western Arctic Ocean in September 2013. The value was as low as 180 μatm in the Chukchi Sea where biological activity was high in early summer. In contrast, $p\mathrm{CO_2}^{\mathrm{sea}}$ in the Canada Basin in September reached 360 μatm, the value comparable to $p\mathrm{CO_2}^{\mathrm{air}}$. Based on the relationship between salinity and TA, we attributed the low salinity water in the Canada Basin mainly to the input of sea-ice melt. Large input of oligotrophic sea-ice melt not only inhibits biological activity, but also facilitates to form a thin surface mixed layer that is easier to reach equilibrium with respect to the atmospheric $\mathrm{CO_2}$. In the area where mixing with riverine output was more dominant than with sea-ice melt, the increase in $p\mathrm{CO_2}^{\mathrm{sea}}$ was indistinct due to the input of riverine nutrients and TA.

In the Canada Basin, $p\mathrm{CO_2}^{\mathrm{sea}}$ was the lowest (~250 μatm) under the surface mixed layer below a strong halocline (difference in density is larger than 2.0 kg m$^{-3}$). This differs from other regions where the lowest $p\mathrm{CO_2}^{\mathrm{sea}}$ was observed in the surface. This subsurface $p\mathrm{CO_2}^{\mathrm{sea}}$ minimum corresponds to PSW and is attributable to the lager net primary production and originally lower DIC of PSW compared to those water of Canada Basin-origin. The subsurface low $p\mathrm{CO_2}^{\mathrm{sea}}$ layer in the Canada Basin has a potential to absorb $\mathrm{CO_2}$ from the atmosphere in the case when it mixes with the surface by a strong turbulance. However, such an increase of $\mathrm{CO_2}$ absorption is unlikely because this stratification is strong enough to resist vertical mixing by wind. Additionally, long-term observations in the Canada Basin suggest that subsurface biological activity has been declining in recent decades.

Our observations could reveal only a part of the complex carbon cycle in the Arctic Ocean. The subsurface $p\mathrm{CO_2}^{\mathrm{sea}}$ minimum is specific to the Canada Basin where circulation of waters generates a complicated water-column structure; the results here are unlikely to be applicable to the entire Arctic Ocean. In the changing Arctic Ocean, in spite of the finding that these subregional variations and processes are essential for better projections of the future carbon cycle, they are not adequately reflected in current models. The areas we can observe in the Arctic Ocean are expected to expand along with the long-term sea-ice retreat. Comprehensive observations are essential especially in such areas because sea-ice melt may cause other effects that are unknown to date.

**Acknowledgement**

We thank the captain, officers, crew and technical staff of R/V *Mirai* for their support on board. This work was financially supported by GRENE Arctic Climate Change Research Project by the Ministry of Education, Culture, Sports, Science and Technology of Japan. We also acknowledge financial support from Japan Society for the Promotion of Science KAKENHI Grant Number 23241002. We appreciate Dr. Wei-Jun Cai and an anonymous reviewer for their valuable review comments.

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

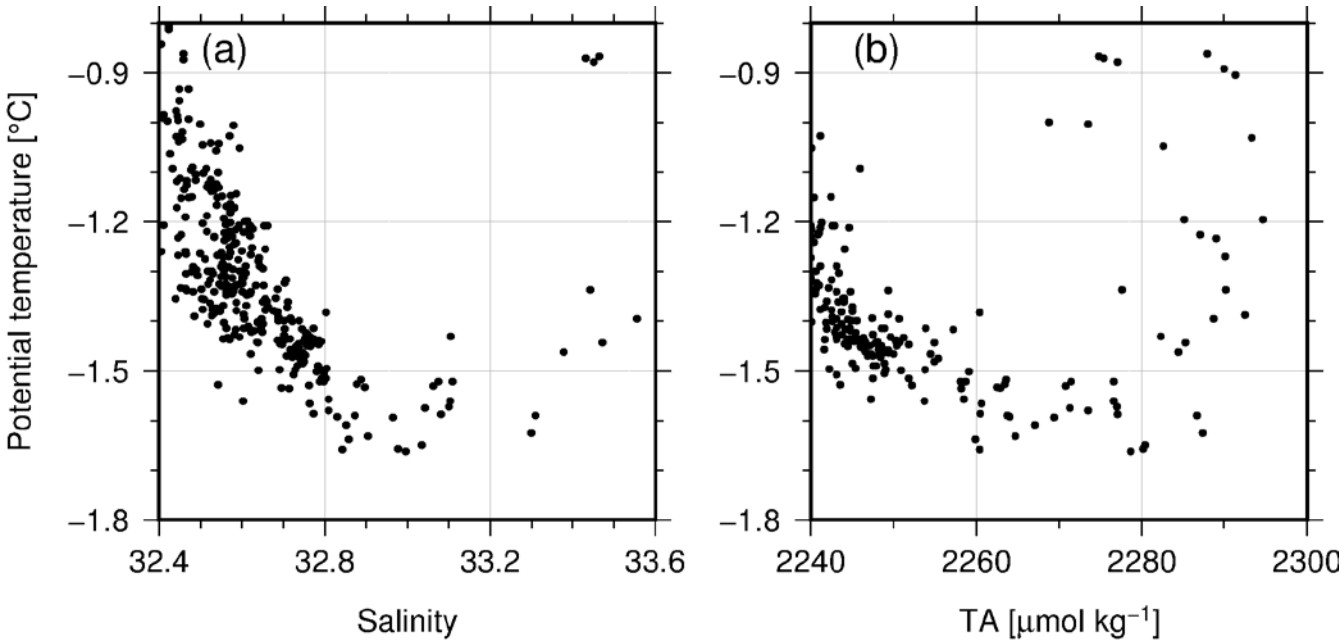

**Figure 1: Water properties around the temperature minimum layer in the Chukchi Sea and the Canada Basin in samples collected**
5    **during cruise MR13-06 from 3 September to 1 October 2013. Potential temperature versus (a) salinity and (b) total alkalinity (TA).**

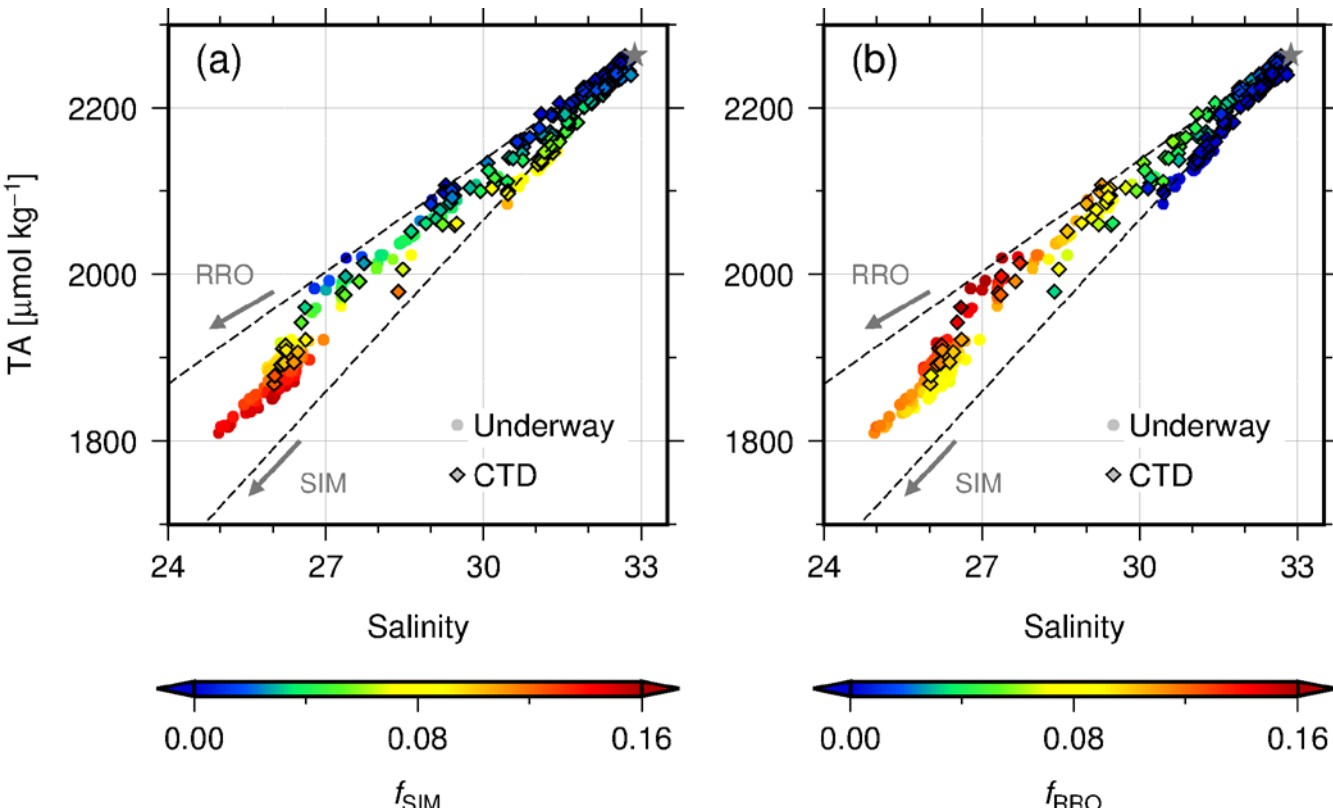

Figure 2: TA versus salinity in the western Arctic Ocean above the temperature minimum layer from 4 to 11 September 2013 color coded for (a) the fraction of sea-ice melt ($f_{SIM}$) and (b) the fraction of riverine outflow ($f_{RRO}$). Circles and diamonds denote surface water and CTD samples, respectively. Star denotes the salinity and TA of Pacific origin water (POW: $S = 32.89$ and TA = 2264.2 μmol kg$^{-1}$). Broken lines extend to the two endmembers, sea-ice melt (SIM: $S = 5$ and TA = 349 μmol kg$^{-1}$) and riverine output (RRO: $S = 0$ and TA = 793 μmol kg$^{-1}$).

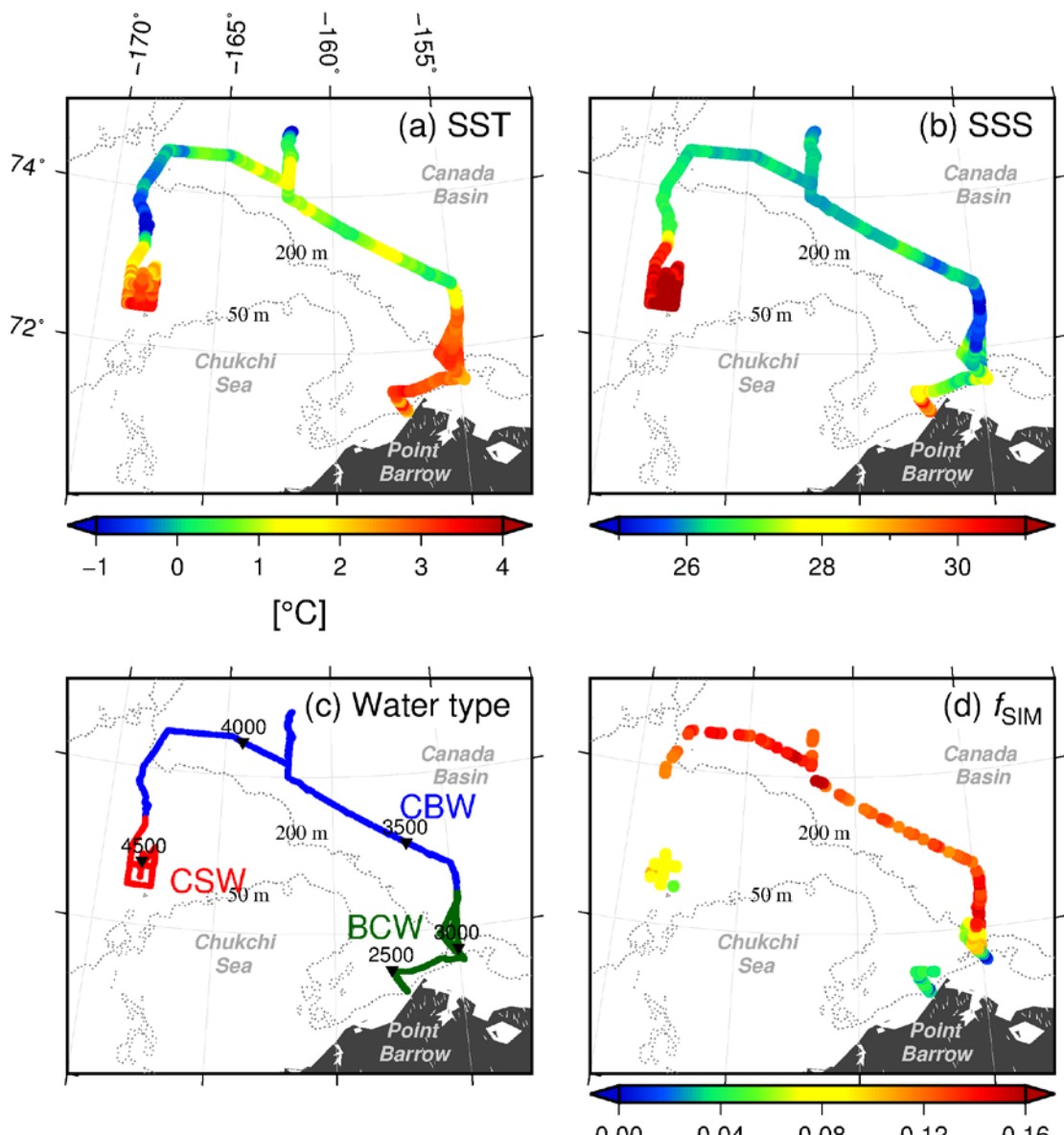

**Figure 3: Surface water properties along the track of cruise MR13-06 from 4 to 11 September 2013. (a) Sea surface temperature (SST), (b) sea surface salinity (SSS), (c) BCW (Barrow Coastal Water), CSW (Chukchi Sea Water), and CBW (Canada Basin Water) water type according to SST and SSS. Number on triangles indicates distance sailed from Dutch harbor, Alaska, USA [km], (d) $f_{SIM}$, (e) $f_{RRO}$, (f) $p$CO$_2^{sea}$, (g) nDIC$_{32}$ = DIC / S * 32; DIC normalized to $S$ =32 and (h) nTA$_{32}$ = TA / S * 32; TA normalized to $S$ =32 . Dotted lines indicate 50 and 200 m isodepths.**

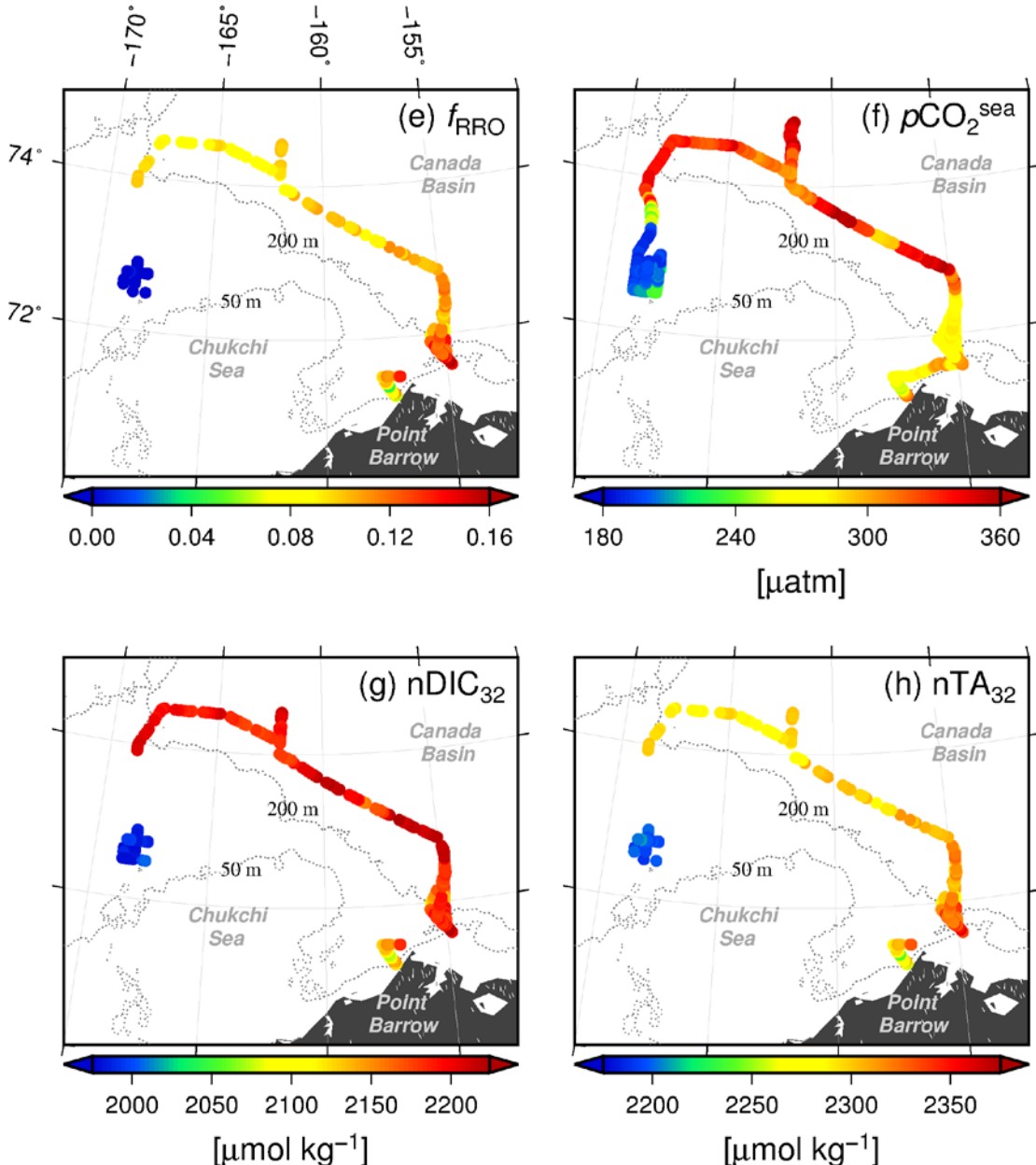

**Figure 3 (continued)**

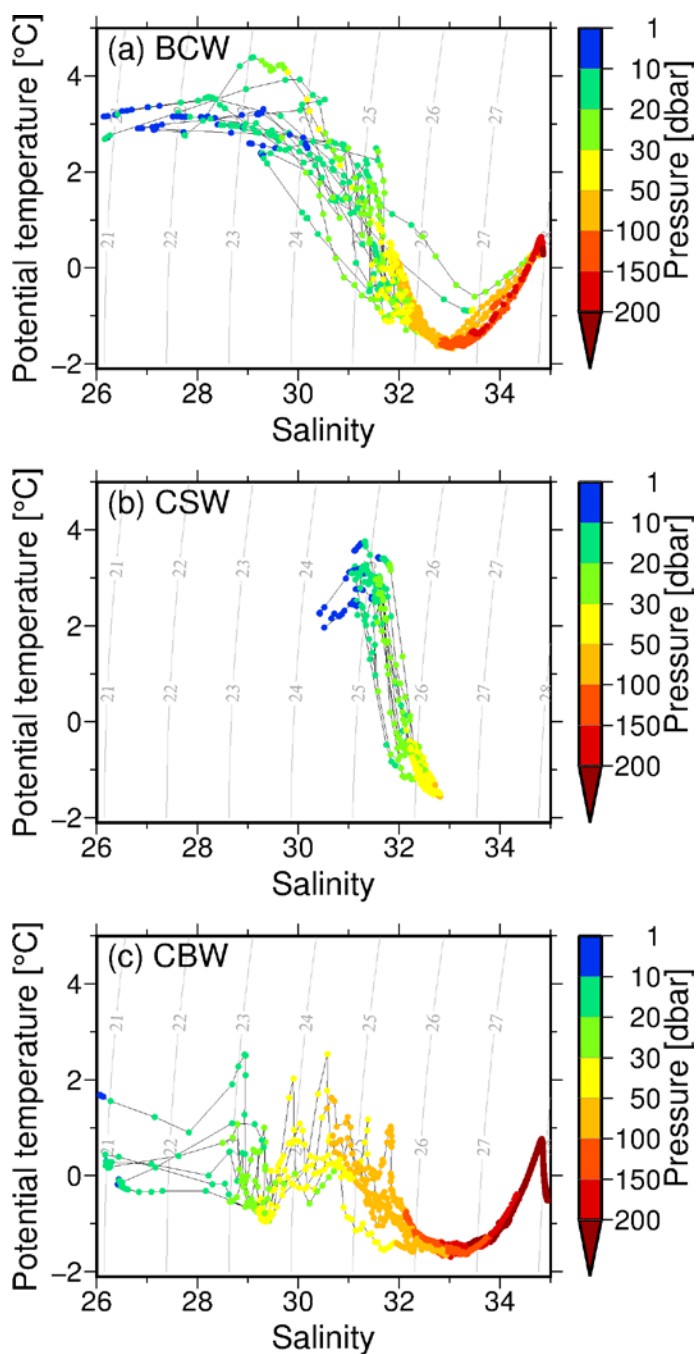

**Figure 4: Column salinity and potential temperature in (a) BCW (Barrow Coastal Water), (b) CSW (Chukchi Sea Water), and (c) CBW (Canada Basin Water). Water pressure is indicated by color. Gray contours indicate potential density ($\sigma_\theta$ = {density – 1} × 1000 [kg m$^{-3}$]).**

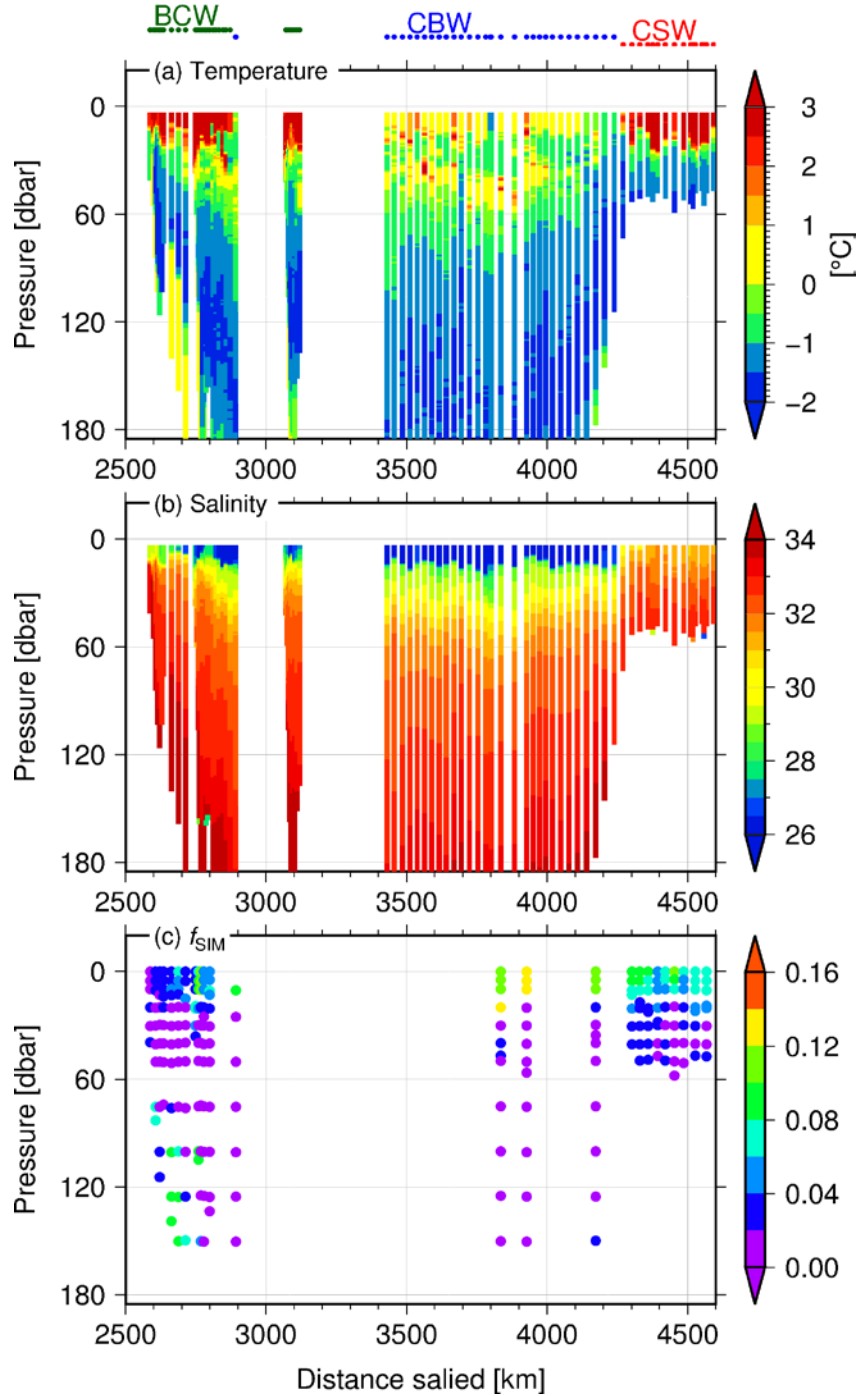

**Figure 5: Water-column profiles of (a) temperature, (b) salinity, (c) $f_{SIM}$, (d) $f_{RRO}$, (e) $pCO_2^{sea}$, and (f) apparent oxygen utilization (AOU) along the cruise track in the period 4–11 September 2013. Data were obtained by CTD and XCTD in (a) and (b), by oxygen sensor SBE43 on CTD in (f), and by discrete bottle samples in (c), (d) and (e). Water types BCW (Barrow Coastal Water, CBW (Canada Basin Water), and CSW (Chukchi Sea Water) are indicated at the top of the figure.**

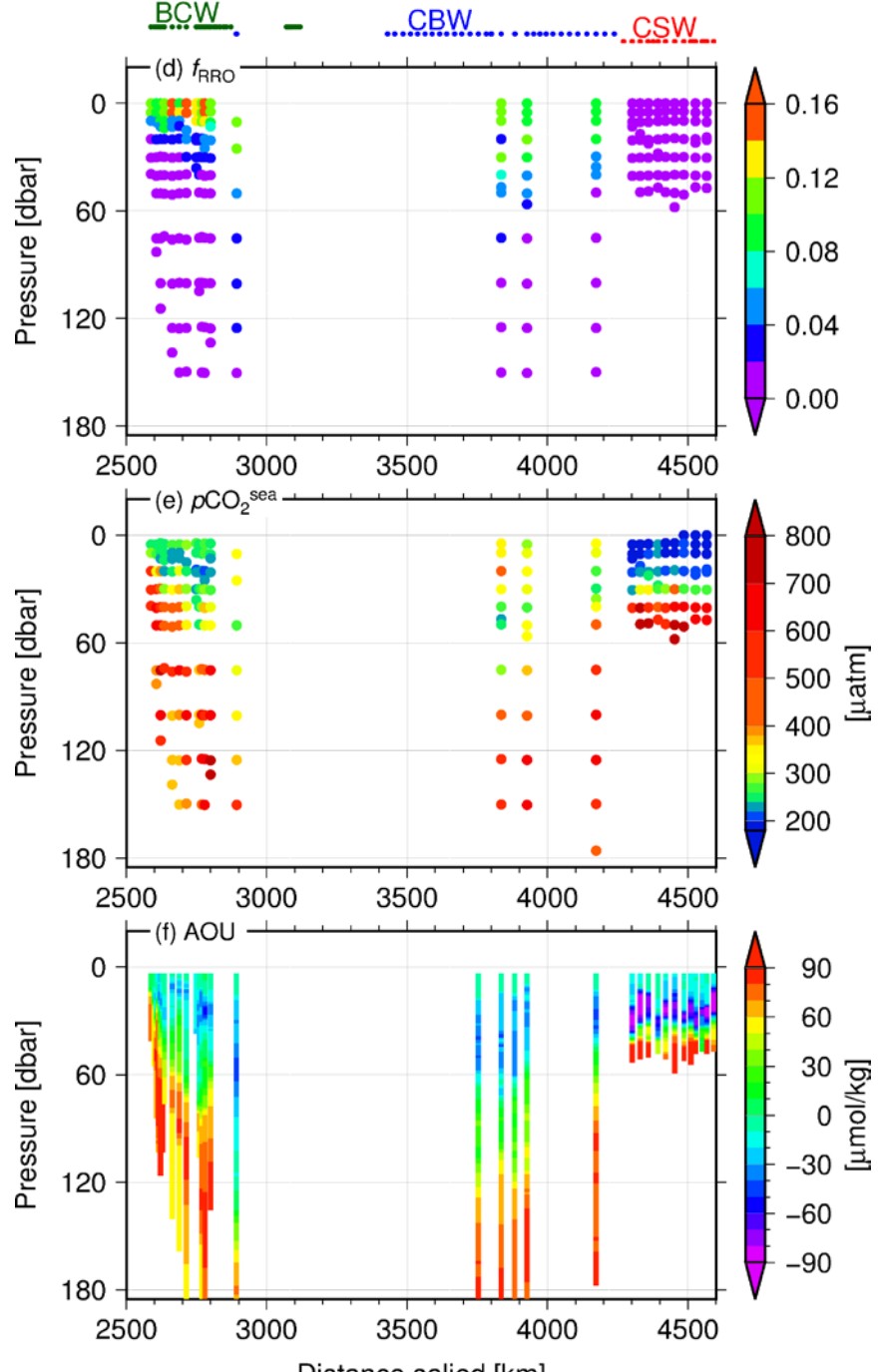

**Figure 5 (continued)**

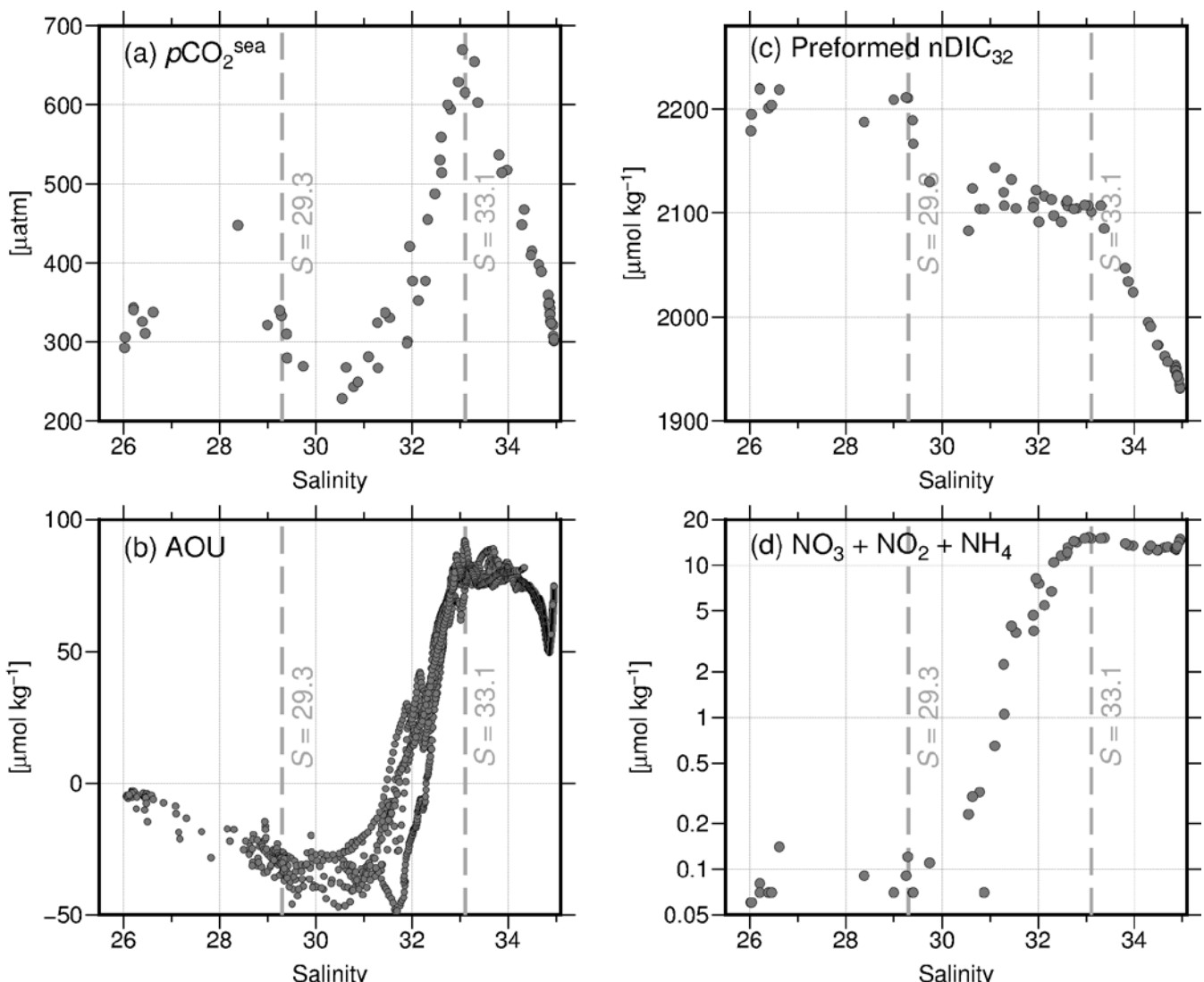

**Figure 6: Variation in several parameters of Canada Basin Water against salinity. (a)** $pCO_2^{sea}$ **in discrete bottle samples, (b) apparent oxygen utilization (AOU) from CTD cast data, (c) preformed nDIC$_{32}$ (= {DIC − AOU * 117 / 170} / S * 32) in discrete bottle samples and dissolved nitrate (NO$_3$ + NO$_2$ + NH$_4$) in logarithmic scale in discrete samples. Salinity of rWML (S = 29.3) and PWW (S = 33.1) were indicated as gray dotted lines.**

**Table 1: Summary of three water types (BCW, Barrow Coastal Water; CBW, Canada Basin Water; and CSW, Chukchi Sea Water) at the surface in the western Arctic Ocean. All samples were pumped up underway from an intake at ship's bottom. Values are averages for samples collected from 4 to 11 September 2013. N denotes the number of samples. $nDIC_{32}$ and $nTA_{32}$ are DIC and TA normalized to $S = 32$ respectively ($nDIC_{32} = DIC / S * 32$; $nTA_{32} = TA / S * 32$). Standard deviation (SD) was listed below each value.**

| Water Type | N | $T$ [°C] | $S$ | DIC [µmol kg⁻¹] | $nDIC_{32}$ [µmol kg⁻¹] | $pCO_2$ [µatm] | TA [µmol kg⁻¹] | $nTA_{32}$ [µmol kg⁻¹] | DIC/TA | $f_{RRO}$ | $f_{SIM}$ |
|---|---|---|---|---|---|---|---|---|---|---|---|
| BCW | 109 | 2.88 | 27.01 | 1827 | 2166 | 274 | 1948 | 2309 | 0.938 | 0.11 | 0.08 |
| SD | | ±0.30 | ±1.37 | ±72 | ±34 | ±13 | ±85 | ±25 | ±0.006 | ±0.02 | ±0.04 |
| CBW | 118 | 0.66 | 26.19 | 1803 | 2203 | 332 | 1882 | 2299 | 0.958 | 0.10 | 0.12 |
| SD | | ±0.58 | ±0.24 | ±19 | ±16 | ±19 | ±16 | ±9 | ±0.004 | ±0.01 | ±0.01 |
| CSW | 54 | 3.03 | 31.06 | 1923 | 1982 | 198 | 2131 | 2196 | 0.903 | -0.01 | 0.08 |
| SD | | ±0.23 | ±0.19 | ±13 | ±6 | ±19 | ±12 | ±3 | ±0.002 | ±0.00 | ±0.01 |