# Peer review of "Low pCO2 under sea-ice melt in the Canada Basin of the western Arctic Ocean"

_Biogeosciences, 2017_

## Referee Comment (RC1) · Anonymous Referee #1 · 5 Jun 2017

Low pCO2 under sea-ice melt in the Canada Basin of the western Arctic Ocean

Naohiro Kosugi, Daisuke Sasano, Masao Ishii, Shigeto Nishino, Hiroshi Uchida, Hisayuki and Yoshikawa-Inoue

Anonymous referee #

General comments

This paper describes results from a single cruise conducted in the autumn of 2013. The hydrography in the region investigated is complex and with large seasonal signatures.

The analysis of water types from salinity and alkalinity provides a convincing picture of

the distribution af water from different sources and is thus a key to the interpretation of the surface and water column data on dissolved oxygen and carbon dioxide.

The primary result reported, as seen from the title of this communication, is a low subsurface pCO2 and negative AOU in the Canada Basin. The authors suggest that this feature, a hidden CO2 sink formed previously in the year, may have significance in the changing Arctic Ocean. Two figures are used to demonstrate this feature, numbers 5 and 6. Strangely, there are much more pCO2 data points for Canada Basin Water in Fig. 6 than in Fig 5e. Fig. 6 indicates a qualitative relation between pCO2 and AOU. This rewiever finds it necessary that to maket he paper acceptable for publication, the authors explore these relations deeper and quanitatively in order to underpin the roles of photosynthesis, respiration and mixing. Also to examine the likely influence of time from formation to observation.

Specific comments

Page Line

3 1 Add information on where cruise started and refer to Fig. 3.

3 8 Is the instrument calibration response linear? What is the estimated uncertainty of measurements reported significantly outside the calibration range?

3 9 Add reference to the "WMO scale".

3 16 Name of author is here Midorikawa but Mirorikawa in the list of references.

3 26 Were the DIC bottles closed after filling?

3 28 The reference (Nippon ANS, Japan) is unsufficient for a description of the instrumentation of extraction/coulometric titration system.

4 4 To which depth were water samples collected?

4 8 Change "same physical conditions" to "same potential temperature and salinity

conditions".

4 10 How were the chlorophyll measurements calibrated?

4 14 The reference (Nippon ANS, Japan) is unsufficient for a description of the instrumentation used for TA measurements.

4 18 Which computation package is used to compute ocean carbonate chemistry?

5 7 Change TARRO to TArro

6 5 Analysis of satellite imaginery is not mentioned in the Data and Methods section. Which data and how processed needs to be added.

6 9 The calculation of CO2 flux and quantification of ?pCO2 reduction needs more detail.

6 20 Change "High" to "Near equilibrium"conditions.

7 10 At what depth lies the NSTM?

7 19 Suggest changing "affected" to "contributed to".

7 31 A figure is needed to illustrate the distinctiveness of the CBW subsurface minima.

14 Fig.1 The placement of this figure as Fig. 1 is strange. It should be after Data and Methods.

15 Fig. 2 The placement of this figure as Fig. 2 is strange. It should be after Data and Methods.

19 Fig.5 Using the calendar date of data collection on the x-axis is unusual. Particularly as there is no date information with the cruise tracks in Fig. 3. Is it possible to use distance sailed instead?

20 Fig. 5f The colour scale does not include the blue observed at lower depths.

Please also note the supplement to this comment:
http://www.biogeosciences-discuss.net/bg-2017-148/bg-2017-148-RC1-
supplement.pdf

―――――――――――――――――――――

---

## Referee Comment (RC2) · Anonymous Referee #2 · 3 Jul 2017

General comments:

The Arctic Ocean is a rapidly changing system that has a highly dynamic CO2 system both seasonally and with the changing physical conditions and climate change scenarios. The authors presented CO2 system data during an autumn of 2013 cruise which will enrich the available CO2 data in the Arctic Ocean and benefit the scientific community (however the dataset is still available via the link). Their results largely support other recent observations that pCO2 is low in ocean margin but high (approaching to the atmospheric pCO2) in the Canada Basin. The explanations they provide are also consistent with other recent publications. Most interesting, the authors observed a subsurface minimal pCO2 structure in the Canada basin. They demonstrated this feature in fig. 5 and discussed the causes for low pCO2 water by analysis the water types, TA-

[Figure]

S and pCO2-AOU relationships. Finally, they discussed the possible fate of this "hidden CO2 sink" and its influence in the future Arctic Ocean (they basically rejected this possibility, which I also agree). I agree with most of their views. Their finding is worthy to be published. However, the main conclusion in this paper is undermined because of not enough data, i.e. low pCO2 in the subsurface of Canada Basin. It is also not clear to me whether pCO2 minimum at 30-50m is due to in situ biological production as they have suggested or subduction of surface water from the highly productive shelf. I think more complete depth profiles from the Niskin bottle samples down to 150-200 m rather than 50m alone from a CTD pumping system will help to elucidate this issue.

Another issue I have with this manuscript is writing. In general they have done a good job in writing except the text around Fig. 5. I don't think the authors put enough thinking into organizing the paper the best they can. One indication is they presented air-sea CO2 flux calculation method in the methods section but never presented air-sea CO2. Did they initially prepared a longer paper and then deleted the flux part? Another indication is in Fig. 5. While it is nice to see the pCO2 minimum with a high frequency depth profile, the depths of such profiles are limited to 50m. However the Pacific winter water and Pacific summer water are all deeper than 50m (if not in this region, they should say it). Thus the entire discussion is not clear. Most important, it is not clear to me whether pCO2 minimum at 30-50m is due to in situ biological production or subduction of surface water from the highly productive shelf. Again, I feel using the depth profile from the Niskin bottle based profiles will help to elucidate this issue. Such depth profiles will also present nutrient profiles to support the argument on nutrient availability (rather than just citing melting pond information).

In summary I'd support the eventual publication of this paper but not at this stage. More data are needed to support their arguments. Specific comments Page 1 Line 23 it is unnecessary to add "e.g." before the citation. Page 2 Line 14 changing "reduces" to "limits" Page3 The pCO2 data set presented in this paper is still not available to readers though the link: (http://www.godac.jamstec.go.jp/darwin/cruise/mirai/mr13-06_leg1/e).

Equation 1 is totally unnecessary. Just cite Takahashi would be enough.

If the authors didn't do any calculation of CO2 flux in this paper, it is totally unnecessary to have the description of air-sea CO2 flux calculation (page 3, line 17- line 25). Line 22, equation (4). I don't recall W92 has a non-zero term. Please check if you cited a more recent Wanninkholf paper and equation. Line 24-25 The wind speed at 24 m height is measured by an anemometer and is extrapolated to 10m. Using an instantaneous wind speed is probably not the best choice for CO2 flux calculation with underway data. The average wind speed from satellite data may make more sense due to equilibrium time for CO2 is pretty long. For example, at 1 pm, if a vessel is at point A where pCO2 is 350 uatm and wind speed is 4 m/s. When the ship arrives at point B at 11 pm the same day where pCO2 is 350 uatm and wind speed increases to 7 m/s. It doesn't make any sense to believe that CO2 uptake flux is much greater at point B than A. If you will use satellite wind, then the fluxes in these two locations are likely the same (that is winds are same for A and B but only changes over a day). However, I must say since calculating flux is not the goal of this paper, this is not a serious problem. Then, of course, there would be no need for the authors to even present the flux calculation equation. Page 4 Line 17 What software or package was used for calculation of carbonate chemistry? Page 5 Line 7 TARRO should be TARRO Line 22-24 The description of " (1) Barrow Coastal Water (BCW) was relatively warm and fresh (SST > 2 °C, SSS < 30.5). (2) Canada Basin Water (CBW) was cold and fresh (SST < 2 °C, SSS < 28). (3) Chukchi Sea Water (CSW) was saline (SSS > 28)" is a little confusing. BCW was fresh SSS<30.5 while CSW was saline (SSS>28). What is the reference for fresh and saline? As I see it (Fig. 3), most of BCW had SSS<28.5 except the very nearshore part while most CSW had SSS>30.5. Only minor clarification is needed here. Page 6 Line 3 removing "the resulting" as low DIC/TA and low pCO2 are the same thing or same result of physical and biogeochemical processes (biological uptake here, but in the basin CO2 evasion from the atmosphere plus strong stratification and low PP in surface water). There is no magic low DIC/TA that leads to low pCO2. Line 9 regarding "half-life of CO2 gas exchange", while I can guess how did you estimate this, it is better to tell

readers. Line 17 Font is different from other context Nutrient in melting pond cannot be a sufficient evidence for limitation of nitrate in surface water. Report directly the nutrient data in water would be better. Line 24 change "pCO2sea" to "pCO2sea " line 29-30 "Reduction in CO2 absorption capacity by riverine discharge was not as large as that by sea-ice melt." This conclusion is not solid. Need more explicit verification. Page 7 Line 4 changing "with depth" to "as depth increases" Line 17-18 "In CSW, the halocline, although not as clear as in the other two subregions, was at almost the same depth." But thermocline is very obvious in CSW (Figure 5a). Line 21 Should "In contrast" be "Likewise"? not clear what is the undertone by this. Page 7 line 19, what is "column variation"? must be water column? Same in lines 26, 29 and 31, all change to "water" column profiles. p.7 Line 26- Page 8 Line 12 The biggest problem here is pCO2 data in CBW is too limited. (only three water column data shown in Figure 5e). Considering the mixing layer structure is complicated in this subregion, it is difficult to see the real pattern. With only 3 station, how to distinguish the real reasons for low pCO2 in subsurface CBW, either due to the local net primary production in CBW or just the water with low pCO2 subducted and advected into the Canada Basin? If the authors could plot the entire water column data (deeper than 50 m in Figure 5), that would provide more information and be helpful to interpret their finding. Also, the Discussion of various waters does not related to Fig. 5 very well, thus causing confusion in reading as the deepest depth is only 50m while the winter water (rWML) is about 120m and summer water (PSW) is even deeper. I am somewhat confused in reading lines 5-13 in p. 8. Since this part is the new point that the authors want to present. It absolutely should be explained very clearly. p.8, line 30, replace "think" with "believe" or "suggest". Table 1 It is not clear whether the average of all the samples were within mixing layer or including the entire water columns. It is probably better to separate the data into mixing layer and below mixing layer for discussion. And please add standard deviation.

---

## Author Comment (AC1) · 31 Jul 2017

First of all, we would like to thank reviewers for their valuable comments on our manuscript. We revised our manuscript carefully by taking these comments into account.

General comments

This paper describes results from a single cruise conducted in the autumn of 2013. The hydrography in the region investigated is complex and with large seasonal signatures. The analysis of water types from salinity and alkalinity provides a convincing picture of the distribution of water from different sources and is thus a key to the interpretation of the surface and water column data on dissolved oxygen and carbon dioxide. The primary result reported, as seen from the title of this communication, is a low subsurface pCO2 and negative AOU in the Canada Basin. The authors suggest that this feature, a hidden CO2 sink formed previously in the year, may have significance in the changing Arctic Ocean. Two figures are used to demonstrate this feature, numbers 5 and 6. Strangely, there are much more pCO2 data points for Canada Basin Water in Fig. 6 than in Fig 5e. Fig. 6 indicates a qualitative relation between pCO2 and AOU. This reviewer finds it necessary that to make the paper acceptable for publication, the authors explore these relations deeper and quantitatively in order to underpin the roles of photosynthesis, respiration and mixing. Also to examine the likely influence of time from formation to observation.

Data have been shown for the entire water column in the Canada Basin in Figure 6a but only for top 60 m in Figure 5e. In Figure 5 in the revised manuscript, we show the data down to the depth of 180 m according to comments by reviewer #2. Figure 6c and 6d were added for quantitative discussion on the subsurface low $p\mathrm{CO_2}$. There were two boundaries in preformed $\mathrm{DIC_{33}}$ and nutrients around $S = 29.3$ and $S = 33.1$. Former and latter corresponded with temperature minimum of rWML and PWW respectively. The layer above $S = 29.3$ was formed in the Chukchi Sea. Water between $S = 29.3$ and $S = 33.1$ can be divided into PSW and PWW by temperature maxiumum around $S = 31$. These were formed in the Chukchi Sea and subducted to the Canada Basin. Low $p\mathrm{CO_2}$ was limited in the upper portion of Pacific origin water, i.e., PSW. This was because DIC in PSW was originally lower, and biological production further reduced it.

Specific comments

Page Line

3 1 Add information on where cruise started and refer to Fig. 3.

The research cruise departed from Dutch Harbor, Alaska on August 31st 2013. This paper focused only a portion of the cruise. This was because general variations in surface $p\mathrm{CO_2}^{\mathrm{sea}}$ in the Western Arctic Ocean have already been well investigated (Bates 2006, Cai et al., 2010). The results from this cruise were not much different from these reports. Therefore, we highlighted water mass characteristic and $\mathrm{CO_2}$ dynamics in the subsurface. Information on the overall cruise and the reason

why we presented the results from a part of the cruise were added to the revised manuscript.

3 8 Is the instrument calibration response linear? What is the estimated uncertainty of measurements reported significantly outside the calibration range?

Response of CRDS we used to the change in the $CO_2$ concentration is practically linear. When we used three standard gases ranging from 206.34 ppmv to 489.28 ppmv for calibration, the residual of each data from linear regression was less than 0.03 ppmv. According to the manufacturer, precision of $CO_2$ measurement above 500 ppmv is 0.1%.

3 9 Add reference to the "WMO scale".

Zhao and Tans (2006) was added as the reference. "WMO scale" was replaced by "WMO X2007 scale" to eliminate ambiguity.

3 16 Name of author is here Midorikawa but Mirorikawa in the list of references.

"Mirorikawa" was changed to "Midorikawa" according to the reviewer's comment.

3 26 Were the DIC bottles closed after filling?

The bottles were always capped with a screw type lid. Bottle filling, transport to measurement system, and discharge were all done through high-density PFA-tubes mounted through the lid.

3 28 The reference (Nippon ANS, Japan) is unsufficient for a description of the instrumentation of extraction/coulometric titration system.

The system was comprised of seawater dispensing unit, a $CO_2$ extraction unit, and a coulometer (Model 3000, Nippon ANS, Inc.). The dispensing unit dispenses the seawater from a glass bottle to a pipette of nominal 15 ml volume. The pipette was kept at $20 \pm 0.05$ °C by a water jacket. Dissolved $CO_2$ in seawater was extracted in a stripping chamber of the $CO_2$ extraction unit by adding 2 cm$^3$ of phosphoric acid (10% v/v). Extracted $CO_2$ was transferred to the coulometer by pure nitrogen gas. Information about the system was added to the revised manuscript.

4 4 To which depth were water samples collected?

Depths of the sample collections in the top 200 m were 0, 5, 10, 20, 30, 40, 50, 75, 100, 125, 150 and 200 m. Samples were also collected at the surface chlorophyll maximum that ranged from 12 m to 92 m. This information was added to the revised manuscript.

4 8 Change "same physical conditions" to "same potential temperature and salinity conditions".

"physical conditions" was changed to "potential temperature and salinity conditions" according to

the comment.

4 10 How were the chlorophyll measurements calibrated?

The instrument was calibrated against pure chlorophyll-a (Sigma-Aldrich Co. LLC.). Description about the calibration was added to the revised manuscript.

4 14 The reference (Nippon ANS, Japan) is unsufficient for a description of the instrumentation used for TA measurements.

Measurement of alkalinity was made using a spectrophotometric system (Nippon ANS, Inc.) based on the scheme of Yao and Byrne (1998). The seawater sampled in the glass bottle is transferred to a sample cell via dispensing unit. The length and volume of the cell are 8 cm and 13 $cm^3$, respectively, and its temperature is kept at 25°C. The TA is calculated by measuring two sets of absorbance at three wavelengths (750, 616 and 444 nm). One is the absorbance of seawater sample before injecting an acid with indicator solution (bromocresol green) and another is the one after the injection of the solution and mixing for 8.5 minutes. Information about system was added to the revised manuscript.

4 18 Which computation package is used to compute ocean carbonate chemistry?

We used for macro package of CO2SYS program for Microsoft Excel (Pierrot et al., 2006). Usage and reference were added.

5 7 Change TARRO to TArro

"TARRO" was changed to "$TA_{RRO}$" according to the reviewer's comment.

6 5 Analysis of satellite imaginary is not mentioned in the Data and Methods section. Which data and how processed needs to be added.

NPP estimation was based on Vertiacal Generalized Production Model by Behrenfeld and Falkowski, (1997). In this method, NPP is estimated from empirical equations. Chlorophyll, sea surface temperature and photosynthetically active radiation obtained by satellite are used as variables. Length of the daytime is also used for the calculation. Description and reference was added to "Measurements and data" section.

6 9 The calculation of CO2 flux and quantification of $\Delta pCO_2$ reduction needs more detail.

At first, initial condition of temperature, salinity, DIC, TA and mixed layer depth was set.

Initial $pCO_2$ ($pCO_2^0$) was calculated. Initial $\Delta pCO_2$ ($\Delta pCO_2^0$) was the difference between $pCO_2^0$ and atmospheric $pCO_2$ ($pCO_2^{air}$).

$$pCO_2^0 = f(T, S, DIC, TA)$$

$$\Delta pCO_2^0 = pCO_2^0 - pCO_2^{air}$$

All parameter except DIC were fixed during the calculation, i.e. evaporation, precipitation and lateral/vertical advection were assumed unchanged. Flux of $CO_2$ ($F_{CO2}$) was calculated from wind speed and gas transfer coefficient. Time step was set to one day. Here, $k$ and $K_0$ denote the solubility of $CO_2$ by Weiss (1974) and gas transfer coefficient by Wanninkhof (2014) respectively.

$$F_{CO2} = kK_0\Delta pCO_2$$

$$k = 0.251 \cdot U_{10}^2 \cdot (Sc/660)^{-0.5}$$

Increase in DIC in each time step was calculated from $F_{CO2}$.

$$\Delta\text{DIC} = \frac{F_{CO2}}{MLD * \rho(T,\ S)}$$

$$DIC_{t+1} = DIC_t + \Delta DIC$$

Here, MLD and $\rho(T,\ S)$ mean mixed layer depth [m] and density of seawater in mixed layer [kg m$^{-3}$] respectively. After each time step, $pCO_2^t$ and $\Delta pCO_2^t$ were calculated from DIC at the time.

$$pCO_{2,t} = f(T, S, DIC_t, TA)$$

$$\Delta pCO_2^t = pCO_2^t - pCO_2^{air}$$

Half-life means the time required to reduce$\Delta pCO_2^t$ to half of $\Delta pCO_2^0$. Description of these processes for calculation of half-life was added to "Calculation" section.

6 20 Change "High" to "Near equilibrium"conditions.

"High" was changed to "Near equilibrium" according to the reviewer's comment.

7 10 At what depth lies the NSTM?

Temperature maximum around $S = 28.8$ in Figure 4c was deemed as NSTM. The depths of NSTM ranged between 15 and 26 m.

7 19 Suggest changing "affected" to "contributed to".

"affected" will be changed to "contributed to" according to the reviewer's comment.

7 31 A figure is needed to illustrate the distinctiveness of the CBW subsurface minima.

Figure 6c and 6d indicated that PSW had lower preformed DIC than surface water and that significant amounts of nutrients remained in PSW. Low $pCO_2$ in PSW was attributable to low preformed DIC and biological production.

14 Fig. 1 The placement of this figure as Fig. 1 is strange. It should be after Data and Methods.

15 Fig. 2 The placement of this figure as Fig. 2 is strange. It should be after Data and Methods.

We divided "Data and Methods" into two new sections, "Measurements and Data" and

"Calculations". These two figures and related descriptions were moved to "Calculations" section.

19 Fig.5 Using the calendar date of data collection on the x-axis is unusual. Particularly as there is no date information with the cruise tracks in Fig. 3. Is it possible to use distance sailed instead?
X-axis of Figure 5 was changed to the distance from the start of cruise. Please see Figure 5 (revised) attached.

20 Fig. 5f The colour scale does not include the blue observed at lower depths.
Color scale changed to cover wider range including positive AOU. Triangles on the color bar were enlarged.

Reference

Bates, N. R. (2006), Air-sea $CO_2$ fluxes and the continental shelf pump of carbon in the Chukchi Sea adjacent to the Arctic Ocean, J. Geophys. Res., 111, C10013, doi:10.1029/2005JC003083.

Behrenfeld, M. J. and Falkowski, P. G., (1997), A consumers guide to phytoplankton primary production models, Limnol. Oceanogr., 42(7), 1479–1491.

Cai, W.-J., Chen, L., Chen, B., Gao, Z., Lee, S.H., Chen, J., Pierrot, D., Sullivan, K., Wang, Y., Hu, X., Huang, W.-J., Zhang, Y., Xu, S., Murata, A., Grebmeier, J. M., Jones, E. P., and Zhang, H. (2010), Decrease in the $CO_2$ uptake capacity in an ice-free Arctic Ocean basin, Science, 329, 556–559, doi:10.1126/science.1189338.

Pierrot, D. E. Lewis, and D. W. R. Wallace. (2006), MS Excel Program Developed for $CO_2$ System Calculations, ORNL/CDIAC-105a. Carbon Dioxide Information Analysis Center, Oak Ridge National Laboratory, U.S. Department of Energy, Oak Ridge, Tennessee. doi: 10.3334/CDIAC/otg.CO2SYS_XLS_CDIAC105a

Yao, W. S. and Byrne, R. H, (1998), Simplified seawater alkalinity analysis: Use of linear array spectrometers, Deep Sea Res., PartI, 45(8), 1383–1392, doi:10.1016/S0967-0637(98)00018-1.

Zhao, C. L., and P. P. Tans, (2006), Estimating uncertainty of the WMO mole fraction scale for carbon dioxide in air, J. Geophys. Res., 111, D08S09, doi:10.1029/2005JD006003.

[Figure]

Figure 5 (revised) Column profiles of (a) temperature, (b) salinity, (c) apparent oxygen utilization (AOU), (d) $pCO_2^{sea}$, (e) $f_{SIM}$, and (f) $f_{RRO}$ along the cruise track in the period 4–11 September 2013. Data were obtained by CTD and XCTD in (a) and (b), by oxygen sensor SBE 43 on CTD in (c), and by discrete bottle samples in (d), (e) and (f). Water types BCW (Barrow Coastal Water, CBW (Canada Basin Water), and CSW (Chukchi Sea Water) are indicated at the top of the figure.

[Figure]

Figure 5 (revised; continued)

[Figure]

Figure 6 (revised) Property-property plots in the Canada Basin Water values for (a) salinity and $p\mathrm{CO}_2^{sea}$ in discrete bottle samples, (b) salinity and apparent oxygen utilization (AOU) from CTD cast data, (c) salinity and preformed DIC ( = {DIC - AOU} / $S$ *32) in discrete bottle samples and (d) salinity and ($\mathrm{NO}_3 + \mathrm{NO}_2 + \mathrm{NH}_4$) in logarithmic scale in discrete bottle samples. Salinity of rWML ($S$ = 29.3) and PWW ($S$ = 33.1) were indicated as gray dotted lines.

---

## Author Comment (AC2) · 31 Jul 2017

First of all, we would like to thank reviewers for their valuable comments on our manuscript. We revised our manuscript carefully by taking these comments into account.

General comments:

The Arctic Ocean is a rapidly changing system that has a highly dynamic CO2 system both seasonally and with the changing physical conditions and climate change scenarios. The authors presented CO2 system data during an autumn of 2013 cruise which will enrich the available CO2 data in the Arctic Ocean and benefit the scientific community (however the dataset is still available via the link). Their results largely support other recent observations that pCO2 is low in ocean margin but high (approaching to the atmospheric pCO2) in the Canada Basin. The explanations they provide are also consistent with other recent publications. Most interesting, the authors observed a subsurface minimal pCO2 structure in the Canada basin. They demonstrated this feature in fig. 5 and discussed the causes for low pCO2 water by analysis the water types, TA-S and pCO2-AOU relationships. Finally, they discussed the possible fate of this "hidden CO2 sink" and its influence in the future Arctic Ocean (they basically rejected this possibility, which I also agree). I agree with most of their views. Their finding is worthy to be published. However, the main conclusion in this paper is undermined because of not enough data, i.e. low pCO2 in the subsurface of Canada Basin. It is also not clear to me whether pCO2 minimum at 30-50m is due to in situ biological production as they have suggested or subduction of surface water from the highly productive shelf. I think more complete depth profiles from the Niskin bottle samples down to 150-200 m rather than 50m alone from a CTD pumping system will help to elucidate this issue.

Sections were expanded down to 180 m in order to cover PSW and PWW. Please see Figure 5 (revised) attached.

Another issue I have with this manuscript is writing. In general they have done a good job in writing except the text around Fig. 5. I don't think the authors put enough thinking into organizing the paper the best they can. One indication is they presented air-sea CO2 flux calculation method in the methods section but never presented air-sea CO2. Did they initially prepared a longer paper and then deleted the flux part?

As you mentioned, calculation of air-sea $CO_2$ flux is not the main theme of this paper. However, we have to mention how to calculate half-life of $\Delta p$CO$_2$. We divided "Data and Methods" into two new sections, "Measurements and Data" and "Calculations". How to calculate the half-life and data used for calculation was described in "Calculations" section.

Another indication is in Fig. 5. While it is nice to see the pCO2 minimum with a high frequency depth profile, the depths of such profiles are limited to 50m. However the Pacific winter water and Pacific summer water are all deeper than 50m (if not in this region, they should say it). Thus the

entire discussion is not clear. Most important, it is not clear to me whether pCO2 minimum at 30-50m is due to in situ biological production or subduction of surface water from the highly productive shelf. Again, I feel using the depth profile from the Niskin bottle based profiles will help to elucidate this issue. Such depth profiles will also present nutrient profiles to support the argument on nutrient availability (rather than just citing melting pond information). In summary I'd support the eventual publication of this paper but not at this stage. More data are needed to support their arguments.

In order to compare water properties, we calculated preformed $nDIC_{32}$ using the following equation.

$$\text{preformed } nDIC_{32} = \frac{DIC - AOU * r_{C:O}}{S} \cdot 32$$

Stoichiometric respiration ratio of $\Delta CO_2/-\Delta O_2$ ($r_{C:O}$) was set to 117/170 (Anderson and Sarminento, 1994). Also concentrations of nutrients measured onboard were used for the analysis. Figure 6c and 6d were added and attached at the end of this reply. Preformed $DIC_{32}$ and dissolved nitrate changed abruptly around $S = 29.3$. Above this layer, dissolved nitrate was almost depleted and preformed $DIC_{32}$ was relatively high (~2200 μmol kg$^{-1}$). Significant nitrates remained and preformed $DIC_{32}$ was low (~2100 μmol kg$^{-1}$) between $S = 29.3$ and $S = 33.1$. $S = 29.3$ corresponds to temperature minimum in the Canada Basin, i.e., rWML (Figure 4c). The layers rWML and above have been formed in the Canada Basin. PSW and PWW which were distributed between rWML and another temperature minimum around $S = 33.1$ were formed in the Chukchi Sea and subducted to the Canada Basin. Low $pCO_2$ (< 300 matm) was mostly limited to the layer just below rWML. This layer corresponds to PSW. Therefore, low $pCO_2$ under the halocline in the Canada Basin was attributable to the subduction of highly productive water from the Chukchi Sea rather than in situ production in the Canada Basin.

Specific comments

Page 1 Line 23 it is unnecessary to add "e.g." before the citation.
We removed "e.g.".

Page 2 Line 14 changing "reduces" to "limits"
We changed "reduces" to "limits".

Page3 The pCO2 data set presented in this paper is still not available to readers though the link: (http://www.godac.jamstec.go.jp/darwin/cruise/mirai/mr13-06_leg1/e).
We have submitted the data to data center of JAMSTEC. However, it has not been uploaded yet. We inquired the data center about it.

Equation 1 is totally unnecessary. Just cite Takahashi would be enough.

If the authors didn't do any calculation of CO2 flux in this paper, it is totally unnecessary to have the description of air-sea CO2 flux calculation (page 3, line 17- line 25).

Calculating regional air-sea $CO_2$ flux was not a goal of our paper. However, it is essential to mention about air-sea $CO_2$ flux since we calculated half-life of $\Delta pCO_2$. Please see subsequent comment about the calculation of half-life.

Line 22, equation (4). I don't recall W92 has a non-zero term. Please check if you cited a more recent Wanninkhof paper and equation.

$k = 0.251 \cdot U_{10}^2 \cdot (Sc/660)^{-0.5}$ was suggested in the most recent Wanninkhof's paper (Wanninkhof 2014). Also Scmidt number was updated to the latest value in Wannikhof (2014).

Line 24-25 The wind speed at 24 m height is measured by an anemometer and is extrapolated to 10m. Using an instantaneous wind speed is probably not the best choice for CO2 flux calculation with underway data. The average wind speed from satellite data may make more sense due to equilibrium time for CO2 is pretty long. For example, at 1 pm, if a vessel is at point A where pCO2 is 350 uatm and wind speed is 4 m/s. When the ship arrives at point B at 11 pm the same day where pCO2 is 350 uatm and wind speed increases to 7 m/s. It doesn't make any sense to believe that CO2 uptake flux is much greater at point B than A. If you will use satellite wind, then the fluxes in these two locations are likely the same (that is winds are same for A and B but only changes over a day). However, I must say since calculating flux is not the goal of this paper, this is not a serious problem. Then, of course, there would be no need for the authors to even present the flux calculation equation.

We used monthly averaged wind speed derived from climate reanalysis JRA-55 (Kobayashi et al., 2015). Wind speed in the Canada Basin in September 2013 ranged 4-5 m sec$^{-1}$. We divided "Data and Methods" into two new sections, "Measurements and Data" and "Calculations". Usage of JRA-55 was added to "Measurement and Data" section. Description about the correction from wind speed at 24 m to that at 10 m was removed.

Page 4 Line 17 What software or package was used for calculation of carbonate chemistry?

We used for macro package of CO2SYS program for Microsoft Excel. Usage and reference (Pierrot et al., 2006) were added.

Page5 Line 7 TARRO should be TARRO

"TARRO" was changed to "TA$_{RRO}$" according to your comment.

Line 22-24 The description of "(1) Barrow Coastal Water (BCW) was relatively warm and fresh

(SST > 2 , SSS < 30.5). (2) Canada Basin Water (CBW) was cold and fresh (SST < 2 C, SSS < 28). (3) Chukchi Sea Water (CSW) was saline (SSS > 28)" is a little confusing. BCW was fresh SSS<30.5 while CSW was saline (SSS>28). What is the reference for fresh and saline? As I see it (Fig. 3), most of BCW had SSS<28.5 except the very nearshore part while most CSW had SSS>30.5. Only minor clarification is needed here.

Difference in water properties between BCW and CBW was remarkable in temperature. On the other hand, that between CBW and CSW was remarkable in salinity. Therefore, we changed Line 21-25 to "We defined three subregions; (1) Barrow Coastal Water (BCW), (2) Canada Basin Water (CBW) and (3) Chukchi Sea Water (3). The boundary between BCW and CBW was 2°C isotherm at 72.5°N, 154.8°E. CBW and CSW was separated 28 psu isohaline at 73.3°N 168.3°E (Fig 3c)."

Page 6 Line 3 removing "the resulting" as low DIC/TA and low pCO2 are the same thing or same result of physical and biogeochemical processes (biological uptake here, but in the basin CO2 evasion from the atmosphere plus strong stratification and low PP in surface water). There is no magic low DIC/TA that leads to low pCO2.

Admittedly, "the resulting" was removed.

Line 9 regarding "half-life of CO2 gas exchange", while I can guess how did you estimate this, it is better to tell readers.

At first, initial condition of temperature, salinity, DIC, TA and mixed layer depth was set.

Initial $pCO_2$ ($pCO_2{}^0$) was calculated. Initial $\Delta pCO_2$ ($\Delta pCO_2{}^0$) was the difference between $pCO_2{}^0$ and atmospheric $pCO_2$ ($pCO_2{}^{air}$).

$$pCO_2{}^0 = f(T, S, DIC, TA)$$
$$\Delta pCO_2^0 = pCO_2^0 - pCO_2^{air}$$

All parameter except DIC were fixed during the calculation, i.e. evaporation, precipitation and lateral/vertical advection were assumed unchanged. Flux of $CO_2$ ($F_{CO2}$) was calculated from wind speed and gas transfer coefficient. Time step was set to one day. Here, $k$ and $K_0$ denote the solubility of $CO_2$ by Weiss (1974) and gas transfer coefficient by Wanninkhof (2014) respectively.

$$F_{CO2} = kK_0\Delta pCO_2$$
$$k = 0.251 \cdot U_{10}^2 \cdot (Sc/660)^{-0.5}$$

Increase in DIC in each time step was calculated from $F_{CO2}$.

$$\Delta DIC = \frac{F_{CO2}}{MLD * \rho(T,\ S)}$$

$$DIC_{t+1} = DIC_t + \Delta DIC$$

Here, MLD and $\rho(T, S)$ mean mixed layer depth [m] and density of seawater in mixed layer [kg m$^{-3}$] respectively. After each time step, $pCO_2{}^t$ and $\Delta pCO_2{}^t$ were calculated from DIC at the time.

$$pCO_{2,t} = f(T, S, DIC_t, TA)$$

$$\Delta pCO_2^t = pCO_2^t - pCO_2^{air}$$

Half-life means the time required to reduce $\Delta pCO_2^t$ to half of $\Delta pCO_2^0$. Description of these processes for calculation of half-life was added to "Calculation" section.

Line 17 Font is different from other context Nutrient in melting pond cannot be a sufficient evidence for limitation of nitrate in surface water. Report directly the nutrient data in water would be better.

We used nutrients data measured on the ship to prepare Figure 6d. In this figure, nutrients depletion (nitrates $< 0.2$ μmol kg$^{-1}$) above $S = 29.3$ (i.e., Canada Basin origin water) and presence of nutrients (about 1 μmol kg$^{-1}$ at $S = 31$) in Pacific origin water are presented.

Line 24 change "pCO2sea" to "pCO2sea "

We changed "$pCO_2$sea" to "$pCO_2^{sea}$".

Line 29-30 "Reduction in CO2 absorption capacity by riverine discharge was not as large as that by sea-ice melt." This conclusion is not solid. Need more explicit verification.

Difference in $pCO_2$ between BCW and CBW was attributable to DIC rather than fraction of fresh water. Please see newly drawn Figure AC1. Relation between $F_{RRO}$ and nDIC$_{32}$ was almost linear in BCW and CSW. However, CBW indicated positive anomaly of nDIC$_{32}$ from linear relation. Additional DIC was imposed on only CBW by possibly air-sea CO$_2$ flux. As a result, "Reduction in CO$_2$ absorption capacity by riverine discharge was not as large as that by sea-ice melt." was partly incorrect. It was replaced by "At the time of the observation, BCW still could absorb more atmospheric CO$_2$ than offshore CBW".

Page 7 Line 4 changing "with depth" to "as depth increases"

We changed "with depth" to "as depth increases".

Line 17-18 "In CSW, the halocline, although not as clear as in the other two subregions, was at almost the same depth." But thermocline is very obvious in CSW (Figure 5a).

In CSW, the halocline, although not as clear as in the other two subregions, was at almost the same depth." was changed to "Unlike the other two subregions, thermocline was more prominent than halocline in CSW."

Line 21 Should "In contrast" be "Likewise"? not clear what is the undertone by this.

"In contrast" was changed to "Likewise".

Page 7 line 19, what is "column variation"? must be water column? Same in lines 26, 29 and 31, all change to "water" column profiles.

All "column profile" were changed to "water column profile".

p.7 Line 26- Page 8 Line 12 The biggest problem here is pCO2 data in CBW is too limited. (only three water column data shown in Figure 5e). Considering the mixing layer structure is complicated in this subregion, it is difficult to see the real pattern. With only 3 station, how to distinguish the real reasons for low pCO2 in subsurface CBW, either due to the local net primary production in CBW or just the water with low pCO2 subducted and advected into the Canada Basin? If the authors could plot the entire water column data (deeper than 50 m in Figure 5), that would provide more information and be helpful to interpret their finding. Also, the Discussion of various waters does not related to Fig. 5 very well, thus causing confusion in reading as the deepest depth is only 50m while the winter water (rWML) is about 120m and summer water (PSW) is even deeper. I am somewhat confused in reading lines 5-13 in p. 8. Since this part is the new point that the authors want to present. It absolutely should be explained very clearly.

Temperature minimum layer around $S = 29.3$ in Figure 5 was rWML. PSW was relatively warm water just below rWML to temperature maximum around $S = 31$. As shown in Figure 6a, $p\mathrm{CO_2}^{sea}$ showed small variability ranging 300-350 µatm in the layers shallower than $S = 29.3$. The lowest $p\mathrm{CO_2}^{sea}$ was seen in the water of $29 < S < 31$. This layer was PSW which was formed in the Chukchi Sea and advected to the Canada Basin. Subsurface minimum in $p\mathrm{CO_2}^{sea}$ in the Canada Basin was formed due to biological production in subducted PSW rather than in Canada Basin origin water. This was supported by low preformed $\mathrm{nDIC_{32}}$ (Figure 6c) and existence of nutrients (Figure 6d) in Pacific origin water between $S = 29.3$ and $S = 33.1$.

p.8, line 30, replace "think" with "believe" or "suggest".
We changed "think" to "suggest".

Table 1 It is not clear whether the average of all the samples were within mixing layer or including the entire water columns. It is probably better to separate the data into mixing layer and below mixing layer for discussion. And please add standard deviation.
All data in Table 1 was for surface water pumped from ship's bottom. This information was added to the caption of Table 1. Standard deviation was added to Table 1 (please see below).

Reference
Anderson, L. A. and J. L. Sarmiento, (1994), Redfield ratios of remineralization determined by nutrient data analysis, Global Biogeochemical Cycles, 8, 65-80.

Kobayashi, S., Y. Ota, Y. Harada, A. Ebita, M. Moriya, H. Onoda, K. Onogi, H. Kamahori, C. Kobayashi, H. Endo, K. Miyaoka, and K. Takahashi, (2015), The JRA-55 Reanalysis: General specifications and basic characteristics. J. Meteor. Soc. Japan, 93, 5-48, doi:10.2151/jmsj.2015-001.

Pierrot, D. E. Lewis, and D. W. R. Wallace. (2006), MS Excel Program Developed for CO2 System Calculations. ORNL/CDIAC-105a. Carbon Dioxide Information Analysis Center, Oak Ridge National Laboratory, U.S. Department of Energy, Oak Ridge, Tennessee. doi: 10.3334/CDIAC/otg.CO2SYS_XLS_CDIAC105a

Wanninkhof, R., (2014), Relationship between wind speed and gas exchange over the ocean revisited, Limnology and Oceanography: Methods, 12, 351-362, doi: 10.4319/lom.2014.12.351.

Table 1 (revised)

Summary of three water types (BCW; Barrow Coastal Water, CBW; Canada Basin Water and CSW; Chukchi SeaWater) at the surface. All sample waters were pumped up underway from an intake at ship's bottom. Values are averages for samples collected from 4 to 11 September 2013. N denotes the number of samples. $nDIC_{32}$ and $nTA_{32}$ are DIC and TA normalized to $S = 32$ respectively ($nDIC_{32} = DIC / S \cdot 32$; $nTA_{32} = TA / S \cdot 32$). Standard deviation (SD) was listed below each value.

| Water Type | N | $T$ [°C] | $S$ | DIC [μmol kg$^{-1}$] | $nDIC_{32}$ [μmol kg$^{-1}$] | $pCO_2$ [μatm] | TA [μmol kg$^{-1}$] | $nTA_{32}$ [μmol kg$^{-1}$] | DIC/TA | $f_{RRO}$ | $f_{SIM}$ |
|---|---|---|---|---|---|---|---|---|---|---|---|
| BCW | 109 | 2.88 | 27.01 | 1827 | 2166 | 274 | 1948 | 2309 | 0.938 | 0.11 | 0.08 |
| (SD) |  | 0.30 | 1.37 | 72 | 34 | 13 | 85 | 25 | 0.006 | 0.02 | 0.04 |
| CBW | 118 | 0.66 | 26.19 | 1803 | 2203 | 332 | 1882 | 2299 | 0.958 | 0.10 | 0.12 |
| (SD) |  | 0.58 | 0.24 | 19 | 16 | 19 | 16 | 9 | 0.004 | 0.01 | 0.01 |
| CSW | 54 | 3.03 | 31.06 | 1923 | 1982 | 198 | 2131 | 2196 | 0.903 | -0.01 | 0.08 |
| (SD) |  | 0.23 | 0.19 | 13 | 6 | 19 | 12 | 3 | 0.002 | 0.00 | 0.01 |

[Figure]

Figure 5 (revised) Column profiles of (a) temperature, (b) salinity, (c) apparent oxygen utilization (AOU), (d) $pCO_2^{sea}$, (e) $f_{SIM}$, and (f) $f_{RRO}$ along the cruise track in the period 4–11 September 2013. Data were obtained by CTD and XCTD in (a) and (b), by oxygen sensor SBE 43 on CTD in (c), and by discrete bottle samples in (d), (e) and (f). Water types BCW (Barrow Coastal Water, CBW (Canada Basin Water), and CSW (Chukchi Sea Water) are indicated at the top of the figure.

[Figure]

Figure 5 (revised; continued)

[Figure]

Figure 6 (revised) Canada Basin Water values for (a) salinity and $p\mathrm{CO_2}^{sea}$ in discrete bottle samples, (b) salinity and apparent oxygen utilization (AOU) from CTD cast data, (c) salinity and preformed $\mathrm{nDIC_{32}}$ ( = {DIC − AOU} / $S \cdot 32$) in discrete bottle samples and (d) salinity and ($\mathrm{NO_3 + NO_2 + NH_4}$) in logarithmic scale in discrete bottle samples. Salinity of rWML ($S$ = 29.3) and PWW ($S$ = 33.1) were indicated as gray dotted lines.

---

## Author Response (AR2)

First of all, we would like to thank reviewers for their valuable comments on our manuscript. We revised our manuscript carefully by taking these comments into account.

*I am glad to see the authors have reorganized the manuscript structure and present the more data in depths deeper than 50m. The data of 0-180m water column are more convincing for reporting the subsurface low pCO2 in the Canada Basin, though data are available at only three stations. I would like to support the publication of this paper with moderate improvement.*

*Main points:*

*p.9, the text below equation 10 should be expanded and made clear. This is the most interesting part of the paper. The authors have improved this argument, but I feel it is still not quite clear. Overall, I agree that the low pCO2 in water of S=29.3-33.1 is partly a result of previous biological removal of CO2 when the water was still in surface during the early part of the summer and partly a continuous biological removal in the subsurface. I think this is what the authors trying to say and I believe this is enough (though say further work is needed to separate the two). I appreciate the authors' effort trying to separate the two, but the approach they took with Preformed nDIC may not be reliable. It is possible that the higher nDIC in the surface is an artifact caused by a non-zero river DIC. In the definition of nDIC one assumes a 0 DIC in the freshwater (e.g., rainwater), which is not the case for the Arctic River. A single endmember normalization (nDIC = DIC/S\*35) will lead to an artificially higher DIC in low salinity water (See Frii et al. paper below). I do not know how serious this would be (you can check nTA to see if your nTA is invariable with salinity). I would prefer you either should do a full mixing model or drop this. However if you choose to keep it, at least admit the potential problem here.*

*Friis, K., A. Kortzinger, and D. W. R. Wallace. 2003. The salinity normalization of marine inorganic carbon chemistry data. Geophys. Res. Lett. 30.*

Both sea ice melt and riverine output contain certain amount of DIC. Therefore salinity normalization causes artificial high DIC in lower salinity. We tried to full mixing model of fresh water endmembers using equation (11).

$$\text{preformed DIC} = f_{POW} \cdot \text{preformed DIC}_{POW} + f_{SIM} \cdot \text{DIC}_{SIM} + f_{RRO} \cdot \text{DIC}_{RRO} \quad (11)$$

DIC in sea ice melt ($\text{DIC}_{SIM}$; $S = 4$) and in riverine output ($\text{DIC}_{RRO}$; $S = 0$) were set to 300 and 800 µmol $\text{kg}^{-1}$ respectively. Preformed DIC in Pacific origin water (preformed $\text{DIC}_{POW}$; $S = 32.89$) was determined to 2170 µmol $\text{kg}^{-1}$ from salinity, DIC and AOU in temperature minimum layer.

In surface layer, DIC observed was almost same as that modeled by Eq. (11). However, DIC observed was lower than that modeled by Eq. (11). This indicated that low preformed DIC in PSW could not be explained by only artifacts arising from salinity normalization with non-zero DIC endmembers.

These model description and results were added to the modified manuscript.

*On gas exchange rate and time.*

*4.2, lines, 8-12, Please also verify if you really calculated half time. It appears to me a 100 days half time is a bit too long in a 15m water depth (See Fig. 4 insert in Cai et al. 2010).*

The reason why half-life indicated in our report seemed so long was the usage of low wind speed. Our report and Cai et al., (2010) used 5 m sec$^{-1}$ and 7 m sec$^{-1}$ as wind speed respectively. Since the gas exchange rate is proportional to the square of the wind speed, our calculation result in about twice length of the half-life as that in Cai et al., (2010).

*4.2 15-18, true, gas exchange is slow, but it was September, nearly 3 months after the ice melt. What maintains the pCO2 at a very low level must also be the relatively high biological production in the CSW as some nutrient is always there in the newly input water from the Pacific in the shelf area there and a relatively deep mixed layer. As the authors recognized that this is in great contrast with those BCW and CBW. Thus this point should be mentioned too. Perhaps, in line 8, modify "This was because net CO2 exchange..." to*

*"This was because of both a relatively still high biological production and slow net CO2 exchange..."*

As indicated, "This was because net CO$_2$ exchange ~" to "This was because of both a relatively still high biological production and slow net CO$_2$ exchange.".

*On data availability*

*Regarding: Most of data used in this paper are available from the JAMSTEC Data Site for Research Cruises (http://www.godac.jamstec.go.jp/darwin/cruise/mirai/mr13-5 06_leg1/e).*

*I noticed that neither pCO2 nor DIC is available online. These are really the main data used in this paper. Please make them available as soon as possible.*

We have submitted all the data used in our report to JAMSTEC data center. We ask them to open the data as soon as possible.

*Abstract: the first part should be shortened as it only confirms what was report before, e.g., Cai et al. (2010) and Elsa et al. 2013. This will leave space to expand the discussion on the subsurface maximum.*

We deleted "We attribute the relatively high $p$CO$_2$$^{sea}$ in the Canada Basin to the shallow mixed layer and limited net primary production." as this result was same as that of Cai et al. (2010) and Else et al., (2013). We added the depth of subsurface low $p$CO$_2$$^{sea}$ layer in the Canada Basin and refereed to oxygen supersaturation in that layer instead.

*Other minor points*
*Page 2 line 6*
*Delete "e.g." before the citation*
We deleted "e.g.".

*Line 14*
*Change "content" to "concentration"*
"Content" was changed to "concentration".

*Page 8 line 29*
*Delete "visited during the cruise"*
We deleted "visited during the cruise".

*Page 9 Line 5*
*should "analyses" be "the analysis"?*
"analyses" was changed to "the analysis"

*Line 9*
*Change "CO2" to "CO2"*
"CO2" was changed to "CO$_2$".
*Page 10 line 30*
*Change "pCO2sea." to "pCO2sea."*
"$p$CO$_2$$^{sea.}$" was changed to "$p$CO$_2$$^{sea.}$".

*Line 10*
   *"Here, our frequent observations facilitated classification of the water masses and their origins; this in turn explains the biological production in the subsurface layer of the Canada Basin."*

*What does "this" refer to? observations? classification of the water masses? Or the subsurface maximum Chla?*

We deleted "this in turn explains the biological production in the subsurface layer of the Canada Basin." as it was not concrete.

*line 26-27*

*Should not use salinity to indicate depths at the same time in your discussion. Easy to get the readers confused.*

*For example, "the water above S=29.3" could be understood as "the water with salinity above (higher than) S=29.3" or "the water above the depth where the salinity is 29.3".*

"$S$ = 29.3" was changed to "the depth where salinity was 29.3".

*Line 18*

*Change "in case it is mixed with" to "in the case when it mixes with"*

"in case it is mixed with" was changed to "in the case when it mixes with".

[revised manuscript text omitted]